# Autophagy impairment in liver CD11c+ cells promotes non-alcoholic fatty liver disease through production of IL-23

Lauriane Galle-Treger[1,7], Doumet Georges Helou [1,7], Christine Quach [1], Emily Howard[1], Benjamin P. Hurrell[1], German R. Aleman Muench[2], Pedram Shafiei-Jahani[1], Jacob D. Painter[1], Andrea Iorga[3,4], Lily Dara[3,4], Juliet Emamaullee [5], Lucy Golden-Mason [3,4,6], Hugo R. Rosen[3,4,6], Pejman Soroosh[2] & Omid Akbari [1,6✉]

There has been a global increase in rates of obesity with a parallel epidemic of non-alcoholic fatty liver disease (NAFLD). Autophagy is an essential mechanism involved in the degradation of cellular material and has an important function in the maintenance of liver homeostasis. Here, we explore the effect of Autophagy-related 5 (Atg5) deficiency in liver CD11c+ cells in mice fed HFD. When compared to control mice, Atg5-deficient CD11c+ mice exhibit increased glucose intolerance and decreased insulin sensitivity when fed HFD. This phenotype is associated with the development of NAFLD. We observe that IL-23 secretion is induced in hepatic CD11c+ myeloid cells following HFD feeding. We demonstrate that both therapeutic and preventative IL-23 blockade alleviates glucose intolerance, insulin resistance and protects against NAFLD development. This study provides insights into the function of autophagy and IL-23 production by hepatic CD11c+ cells in NAFLD pathogenesis and suggests potential therapeutic targets.

[1] Department of Molecular Microbiology and Immunology, Keck School of Medicine, University of Southern California, Los Angeles, CA, USA. [2] Janssen Research and Development, San Diego, CA, USA. [3] Division of Gastrointestinal and Liver Diseases, Department of Medicine, Keck School of Medicine, University of Southern California, Los Angeles, CA, USA. [4] Research Center for Liver Disease, Keck School of Medicine, University of Southern California, Los Angeles, CA, USA. [5] Department of Surgery, Keck School of Medicine, University of Southern California, Los Angeles, CA, USA. [6] Department of Medicine, Keck School of Medicine, University of Southern California, Los Angeles, CA, USA. [7] These authors contributed equally: Lauriane Galle-Treger, Doumet Georges Helou. ✉email: akbari@usc.edu

Nonalcoholic fatty liver disease (NAFLD) is the most common form of chronic liver disease in the United States, affecting about 30% of the population and is becoming increasingly widespread because of the rising prevalence of obesity worldwide. Indeed, NAFLD is often associated with metabolic and cardiovascular disorders, such as obesity, insulin resistance, hypertension, dyslipidemia, and type 2 diabetes. NAFLD, which is reversible, is characterized by excess fat deposits in the liver (steatosis) and can subsequently lead to the more aggressive nonalcoholic steatohepatitis (NASH), which is associated with hepatic inflammation leading to apoptosis, fibrosis, and eventually cirrhosis, and hepatocellular carcinoma (HCC)[1,2]. Elucidating the mechanisms involved in the development of NAFLD is crucial in developing effective treatments for patients and discovering preventive methods for individuals at high risk for NAFLD and other potentially serious complications such as HCC.

Autophagy is the intracellular process responsible for the regulated recycling of damaged organelles and the degradation of cytosolic materials. Targeted cytoplasmic components are sequestered by double-membrane vesicles named autophagosomes, which then fuse with lysosomes to release enzymes that degrade the content of the merged vesicles[3–6]. Autophagy-related 5 (Atg5) is a key protein involved in the extension of the phagophore membrane in autophagic vesicles and has a critical function in the autophagy mechanism. Previous studies have shown that dysregulation of autophagy can regulate lipid accumulation, injury, and inflammation, leading to fibrosis and carcinogenesis in NAFLD[7]. Indeed, hepatic autophagy is known to be impaired in NAFLD[8–11]. Thus far, the specific activity of autophagy in hepatic myeloid cells during NAFLD development has not been clearly defined.

Prior studies have shown that high-fat diet (HFD) treatment induces the expression of CD11c in myeloid cells[12–14]. Interleukin-23 (IL-23) is a heterodimeric cytokine comprising the IL-12p40 and IL-23p19 subunits and has been well documented as a critical driver of multiple inflammatory conditions[15]. Notably, our group has shown that deletion of autophagy in CD11c+ dendritic cells (DCs) was associated with increased levels of pro-inflammatory cytokine secretion, particularly IL-23[16]. We analyzed a previously published transcriptome analysis of livers from patients with advanced NAFLD compared to mild NAFLD. The results suggest that disease severity is associated with decreased expression of genes related to autophagy and increased activation of the IL-23 pathway genes[17].

Here, we analyze the activity of autophagy in hepatic myeloid cells using a model of *Atg5* deletion in CD11c+ cells, Atg5 CD11cKO mice which develop NAFLD and insulin resistance when administered a high-fat diet. In addition, Atg5 CD11cKO mice have increased IL-23 secretion in hepatic CD11c+ myeloid cells. Utilizing a neutralizing antibody against IL-23[18,19], we demonstrate that preventative inhibition of IL-23 not only protects against the development of glucose intolerance and insulin insensitivity, but could also dampen the development of NAFLD. Interestingly, we further demonstrate that therapeutic anti-IL-23 treatment could ameliorate the established glucose homeostasis impairment and prevent the development of NAFLD. Our findings provide insights on the function of IL-23 induction in NAFLD development and introduce IL-23 as a potential therapeutic target in NAFLD.

## Results

**NAFLD patients have decreased autophagy and increased IL-23 pathway.** We extracted and reanalyzed a liver transcriptome dataset from a previously published study with 40 mild NAFLD and 32 advanced NAFLD patients (Fig. 1a)[17]. The genes are shown as their moderated t-statistic value which is calculated based on the fold change (FC) between patients with advanced NAFLD compared to mild cases, as defined by the NASH activity score (NAS). Interestingly, we observed that the expression of autophagy-related genes such as *BECN1*, *GABARAP*, *RAB7A*, *MAP1LC3*, *ATG5*, and *RAB7A* was decreased in livers of advanced NAFLD patients. These genes are critical for autophagy and are involved in different steps of the autophagy process. Indeed, *BECN1* is important for autophagy initiation; *RAB7A*, *MAP1LC3*, *ATG5* are crucial for autophagosome membrane elongation whereas *RAB7A* is involved in autophagosome trafficking and fusion with lysosomes. We also observed that the expression of genes associated with the IL-23 pathway was induced in livers of patients with advanced NAFLD (Fig. 1a). IL-23 activity in inflammation is associated with a unique gene signature that includes *IL17A*, *IL23A*, *CCL20*, and *IL1R1*. These gene expressions were notably increased in the livers of patients with advanced NAFLD. In parallel, we also extracted and reanalyzed a liver transcriptome dataset from another previously published study on mice fed either a normal chow diet (NCD) or a 40% HFD for five weeks (Fig. 1b)[20]. Genes are presented as expression FC values between NCD and HFD. Similar to what we observed in human liver samples, autophagy-related genes such as *Atg5*, *Gabarap*, *Atg13*, *Bnip*, *Map1lc3a*, and *Becn1* were reduced in the livers of mice fed HFD. Genes such as *Il22*, *Il17ra*, *Il17rb*, and *Il23a* involved in the IL-23 pathway were induced in samples derived from HFD mice. These results validate the mouse HFD model as a clinically relevant early translational model to study the pathogenesis and mechanisms of human NAFLD.

**HFD induces CD11c and represses autophagy in hepatic CD11c+ cells.** Intriguingly, the expression of *ITGAX* which codes for the CD11c integrin was also highly induced in liver samples from advanced NAFLD patients (Fig. 1a). Traditionally, CD11c expression is considered a marker of DCs, however, studies have established that CD11c expression is also induced in lipid-rich conditions in myeloid cells[12–14]. To confirm our observation that CD11c expression is affected by high-lipid content, we fed NCD or HFD to C57BL/6 mice for 14 weeks and assessed CD11c expression by flow cytometry on live hepatic CD45+ cells (Fig. 2a). In response to the HFD treatment, we observed that CD11c expression was strongly induced in hepatic immune cells with the percentage of hepatic CD45+ CD11c+ cells increasing from 18 to 34%. This result is consistent with our previous observation that *ITGAX* expression was increased in the livers of patients with advanced NAFLD[17].

Next, we assessed the effect of HFD exposure on autophagic flux in CD11c+ cells isolated from the livers of C57BL/6 mice. We measured the conversion of LC3-I to LC3-II and p62 protein expression by Western Blot (Fig. 2b). Upon autophagy initiation, LC3-I is converted by a protein complex into the LC3-II form. LC3-II is a well-characterized protein that localizes to the autophagosome membrane during the autophagy process from phagophore to lysosomal degradation. The p62 protein, a cargo adapter, interacts with autophagic substrates and delivers them to autophagosomes for degradation. In the process, p62 itself is degraded and when autophagy is induced, a corresponding decrease in p62 levels is observed. Our results showed that CD11c+ cells isolated from mice fed HFD had decreased levels of LC3-I to LC3-II conversion and high accumulation of p62, as compared to mice fed NCD (Fig. 2b). The data in Fig. 1b do not show an upregulation in the transcript levels of p62 encoding gene (*Sqstm1*) in the livers of HFD-fed mice, suggesting that p62 accumulation is due to a defect in protein degradation rather than

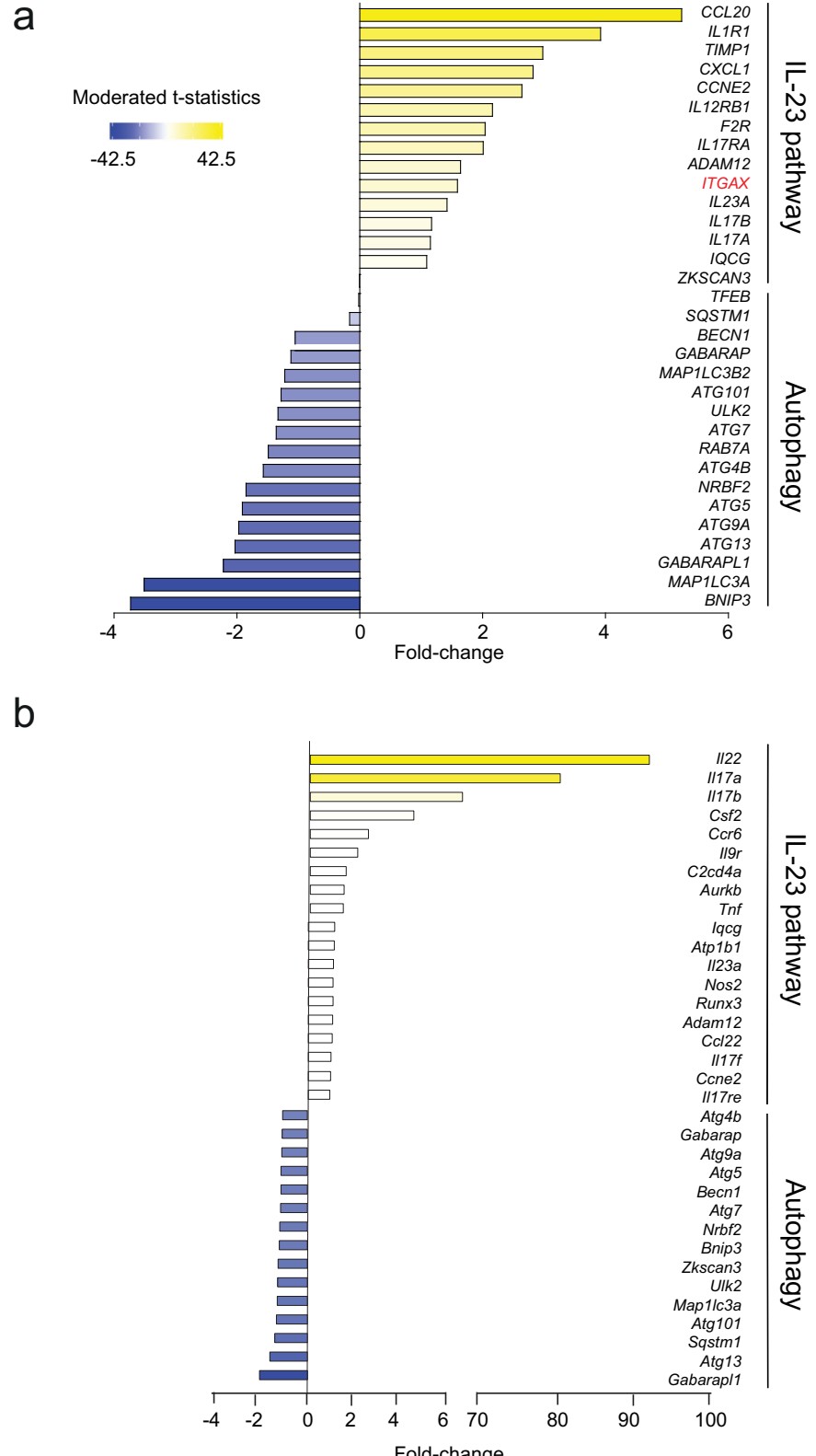

**Fig. 1 Autophagy genes are inhibited in human and mouse livers in lipid-rich contexts. a** Color-coded histogram of the expression levels of genes associated with the IL-23 pathway or the autophagy mechanism. The genes are shown as their moderated *t*-statistic value which is calculated based on the fold-change between patients with advanced NAFLD compared to mild cases. Transcriptome data was obtained and reanalyzed for 72 patients (40 patients with mild NAFLD and 32 patients with advanced NAFLD). **b** Color-coded histogram of the expression levels of genes associated with the IL-23 pathway or the autophagy mechanism. The genes are shown as their fold-change values between WT mice fed NCD or HFD for 7 weeks (*n* = 2).

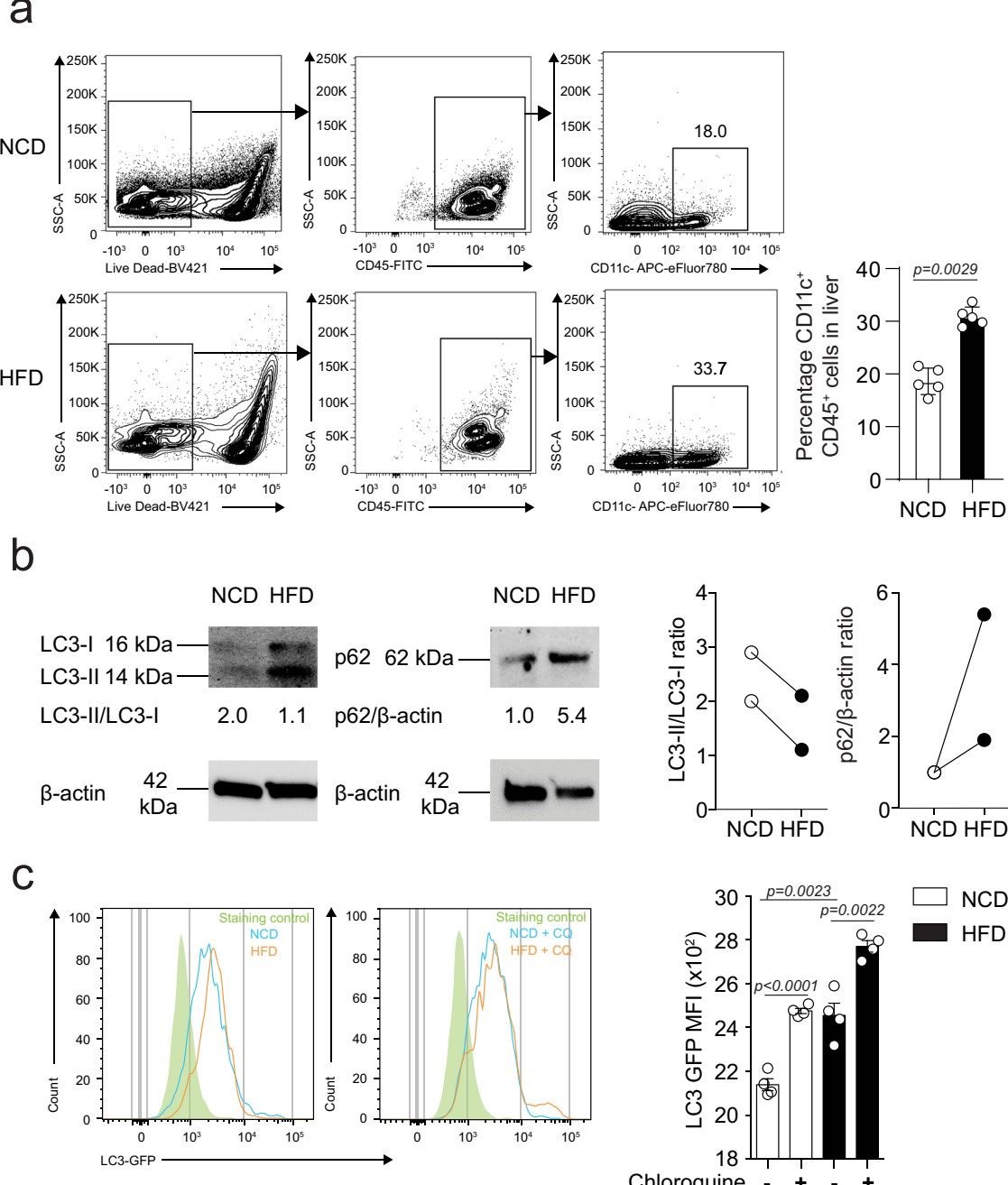

**Fig. 2 HFD induces CD11c expression and impairs the autophagic flux in hepatic CD11c+ cells. a** Gating strategy of hepatic CD11c+ CD45+ cells isolated from C57BL/6 mice fed NCD or HFD (left panel). Percentage of hepatic CD11c+ CD45+ cells (right panel), $n = 5$ mice. **b** Western blot analysis of LC3-I/II and p62 levels in hepatic CD11c+ cells isolated from C57BL/6 mice (NCD vs. HFD) (left panel showing LC3 bands and a middle panel showing p62 bands). Densitometric quantification of LC3II/LC3I ratio (normalized to β-actin) and p62/β-actin (right panel), $n = 2$ independent experiments. **c** Flow cytometry quantification of LC3 expression in hepatic CD11c+ cells isolated from C57BL/6 mice (NCD vs. HFD) and cultured for 15 h in the presence or absence of chloroquine (CQ at 25 μM), $n = 4$ mice. Error bars are the mean ± SD, two-tailed Student's $t$ test. Experiments were performed at least two times. Source data are provided as a Source Data file.

transcriptional upregulation. To check the effect of HFD on LC3 degradation, we treated CD11c+ cells isolated from the livers of LC3-GFP mice (NCD vs. HFD) with chloroquine (CQ), an autophagy inhibitor, and measured by flow cytometry the intensity of the GFP signal. Without the CQ treatment, hepatic CD11c+ cells isolated from mice fed HFD had an elevated baseline accumulation of total LC3 protein compared to cells isolated from mice fed NCD. Interestingly, the total LC3 protein

levels were even higher after CQ treatment confirming that autophagy is inhibited in CD11c+ cells under HFD conditions and lysosomal degradation is disrupted (Fig. 2c). Since autophagy is a dynamic process, we also treated hepatic CD11c+ cells isolated from C57BL/6 mice fed NCD or HFD with CQ and quantified LC3 conversion and p62 protein levels by Western Blot (Supplementary Fig. 1a, b). Our results showed that in the presence of CQ, LC3-I to LC3-II conversion was highly impaired

in the hepatic CD11c$^+$ cells from WT mice fed HFD, in association with an aberrant p62 accumulation. Collectively, these data confirm a diminished autophagic degradation activity in hepatic CD11c$^+$ cells under HFD conditions.

**DCs accumulate in the livers of Atg5 CD11c$^{KO}$ mice fed HFD.** Following the observations that HFD treatment induced CD11c expression and also impaired autophagic flux in hepatic CD11c$^+$ cells, we next conditionally deleted Atg5 in CD11c$^+$ cells (Atg5 CD11c$^{KO}$ mice) and used floxed littermate wild-type (WT) mice as controls. We first confirmed the efficiency and the specificity of the *Atg5* deletion by RT-qPCR in Atg5 CD11c$^{KO}$ mice fed NCD. As previously described[16], Atg5 CD11c$^{KO}$ mice do not express Atg5 in hepatic CD11c$^+$ cells, and the expression of Atg5 is unchanged in hepatic CD11c$^-$ cells as compared to WT mice (Supplementary Fig. 1c). After 14 weeks of NCD or HFD feeding, CD11c expression was assessed in the different hepatic myeloid subpopulations. Hepatic myeloid cells are highly heterogeneous, and their different subsets are still being characterized. In our gating strategy, neutrophils are excluded according to their high side scatter (Supplementary Fig. 1d) and monocytes are identified as Ly6C$^{hi}$CD11b$^+$ cells on gated live CD45$^+$ cells. From Ly6C$^-$ populations, macrophages are gated as CD64 and F4/80 double-positive cells. Recently, multiple populations of liver macrophages including TIM4$^+$CLEC4f$^+$ resident Kupffer cells (TIM4$^+$ KC), TIM4$^-$CLEC4f$^+$ monocyte-derived Kupffer cells (TIM4$^-$ KCs) and CLEC4f$^-$ macrophages have been identified (Fig. 3a)[21]. Therefore, we examined if these macrophage subpopulations were expressing CD11c and how *Atg5* deletion affected the expression of the CD11c integrin in mice fed NCD or HFD (Fig. 3b, c). We also examined the effect of autophagy deletion on DCs, identified as CD11c and MHC-II double-positive cells. We observed that HFD increases CD11c expression in the CLEC4f$^-$ macrophage subpopulation of WT mice, while Atg5 CD11c$^{KO}$ mice display high expression under both NCD and HFD conditions. In parallel, there was no difference in the levels of CD11c expression in TIM4$^-$ and TIM4$^+$ KC subpopulations in WT and Atg5 CD11c$^{KO}$ mice fed NCD or HFD (Fig. 3b). Next, we measured the percentages of the hepatic myeloid subpopulations in the four different mouse groups (Fig. 3c). Feeding HFD increased the percentages of CLEC4f$^-$ macrophages and TIM4$^-$ KC in WT and Atg5 CD11c$^{KO}$ mice, however, the TIM4$^+$ KC subpopulation that displays a relatively low expression of CD11c was decreased in both groups. Interestingly, the percentages of DCs (CD11c$^+$ MCH-II$^+$) were significantly increased only in Atg5 CD11c$^{KO}$ mice fed HFD (Fig. 3c).

In parallel, we did not notice any effect of *Atg5* deletion on percentages of CD11c$^+$ cells in visceral adipose tissue (VAT) or spleen in NCD-fed Atg5 CD11c$^{KO}$ mice (Supplementary Fig. 1e–h). Similarly, we observed no effect of *Atg5* deletion on percentages of VAT myeloid cells in HFD-fed Atg5 CD11c$^{KO}$ mice (Supplementary Fig. 2a, b). Altogether these results demonstrate that regardless of the mice genotype, HFD is associated with an accumulation of recruited macrophage population (TIM4$^-$ KC, and CLEC4f$^-$ macrophages) expressing high levels of CD11c. Additionally, HFD results in the loss of resident macrophages (TIM4$^+$ KC) expressing low levels of CD11c, while HFD feeding in Atg5 CD11c$^{KO}$ remarkably induces DC accumulation.

**Atg5 CD11c$^{KO}$ mice fed HFD develop high insulin resistance and NAFLD.** To study the effect of autophagy deletion in a lipid-rich context, WT and Atg5 CD11c$^{KO}$ mice were fed HFD for 14 weeks. NAFLD development is associated with comorbidities

including obesity, insulin resistance, and Type 2 diabetes. Based on these principles, we examined a variety of metabolic parameters to assess whether deletion of autophagy in CD11c$^+$ cells would alter the development of insulin resistance (Fig. 4a). We observed that both groups had increased total weights and fasting blood glucose levels in response to the HFD treatment, however, Atg5 CD11c$^{KO}$ mice had higher total weights and fasting glucose levels when compared to controls (Fig. 4b, c). We also measured the serum insulin levels in control and Atg5 CD11c$^{KO}$ mice after 14 weeks of HFD. Atg5 CD11c$^{KO}$ mice had serum insulin levels twice as high as concentrations observed in WT mice (Fig. 4d). Glucose tolerance and insulin sensitivity were also tested and quantified at 14 weeks (Fig. 4e, f). Atg5 CD11c$^{KO}$ mice were less tolerant to glucose and less sensitive to insulin with increased area under the curve (AUC) for both tests. These results indicate that deletion of *Atg5* in CD11c$^+$ cells induces insulin resistance.

After 14 weeks of exposure to HFD, mice were euthanized. Explanted livers from Atg5 CD11c$^{KO}$ mice were pale and twice as heavy as control livers, highlighting the development of hepatomegaly and suggesting steatosis (Fig. 4g, h). Hepatomegaly in Atg5 CD11c$^{KO}$ mice could not be attributed to total body weight gain after HFD, as the liver weight to total body weight ratio was significantly higher in Atg5 CD11c$^{KO}$ mice when compared to WT controls (Supplementary Fig. 3a). Atg5 CD11c$^{KO}$ mice also had increased VAT weight compared to WT controls (Supplementary Fig. 3b). We phenotyped Atg5 CD11c$^{KO}$ and WT mice with metabolic cages and analyzed their body composition by nuclear magnetic resonance (NMR) spectroscopy. Consistently with our previous observation Atg5 CD11c$^{KO}$ mice had a higher percentage of fat mass (Supplementary Fig. 3c). Although we observed no difference in energy expenditure, Atg5 CD11c$^{KO}$ mice had increased food intake and were more active at night during feeding time (Supplementary Fig. 3d–f). To confirm that the observed phenotypic differences were not due to a difference in microbiota, we co-housed Atg5 CD11c$^{KO}$ and WT mice together in the same cages. After 14 weeks of HFD, the phenotypic differences such as increased total weight, liver/total weight ratio, VAT weight, and fasting blood glucose in Atg5 CD11c$^{KO}$ mice remained the same (Supplementary Fig. 3g–j).

We also assessed the metabolic parameters in Atg5 CD11c$^{KO}$ and WT mice fed NCD for 14 weeks. We did not note any difference in total weight, liver weight, fasting blood glucose, and ALT levels between groups (Supplementary Fig. 3k–n). Overall, these observations demonstrate that the described metabolic differences are independent of the microbiota but are induced after chronic HFD exposure.

Next, we measured serum and liver triglycerides (TG) levels (Fig. 4i) and determined that Atg5 CD11c$^{KO}$ mice had higher circulating and hepatic concentrations of TG. To explore if the hepatomegaly and the abnormal TG accumulation in the liver of Atg5 CD11c$^{KO}$ mice resulted in liver inflammation, we quantified the serum alanine aminotransferase levels (ALT). ALT levels were increased in hepatic CD11c$^+$ cells in Atg5 CD11c$^{KO}$ mice when compared to controls ($p < 0.05$, Fig. 4j). Histological analysis was conducted to evaluate steatosis, ballooning, lobular inflammation, and NAS of control and Atg5 CD11c$^{KO}$ mice (Fig. 4k, l). After comparison between the two groups, we identified that Atg5 CD11c$^{KO}$ mice had more steatosis, ballooning, and lobular inflammation compared to WT mice resulting in an increased NAS score. These observations are consistent with the hematoxylin and eosin (HE) staining shown in Fig. 4k. These data establish that deletion of *Atg5* in CD11c$^+$ cells in mice induces hepatomegaly and the development of NAFLD leading to liver dysfunction.

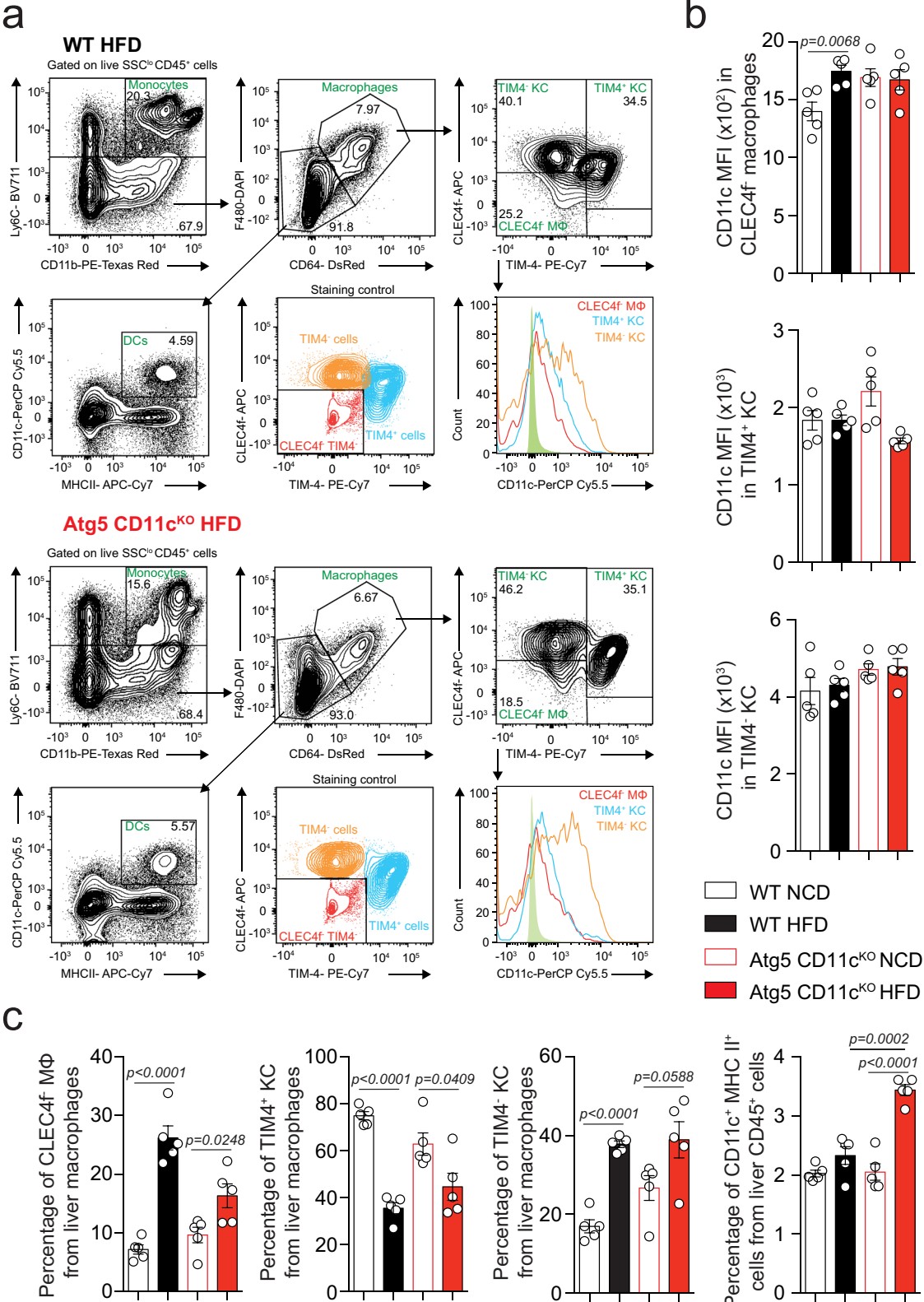

**Fig. 3 Loss of Atg5 increases dendritic cell accumulation in mice fed HFD. a** Gating strategy was used to identify myeloid populations in mice fed NCD or HFD for 14 weeks (top panel WT cells, bottom panel Atg5 CD11c$^{KO}$ mice). The level of isotype-matched stain defines TIM4$^-$ CLEC4f$^-$ population and discriminates TIM4$^+$ and TIM4$^-$ cells. **b** Histogram of CD11c expression in macrophage subpopulations (CLEC4f$^-$ macrophages, TIM4$^+$ KC and TIM4$^-$ KC) from WT and Atg5 CD11c$^{KO}$ mice. **c** Percentage of CLEC4f$^-$ macrophages, TIM4$^+$ KC and TIM4$^-$ KC from total liver macrophages in WT and Atg5 CD11c$^{KO}$ mice (NCD vs. HFD). Error bars are the mean ± SD, ns nonsignificant, two-tailed Student's $t$ test, $n = 5$ mice. Experiments were performed three times. Source data are provided as a Source Data file.

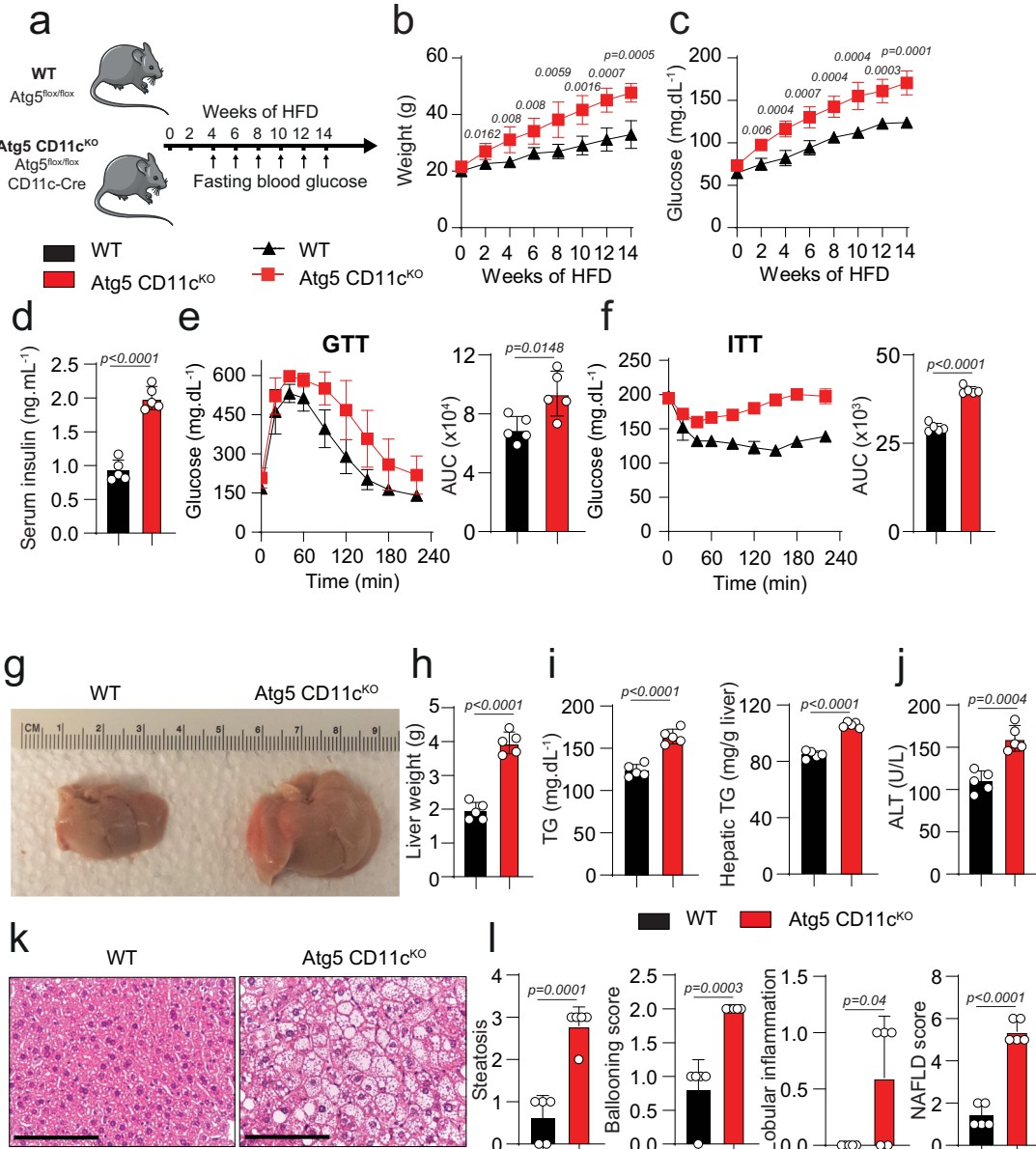

**Fig. 4 *Atg5* deletion in CD11c⁺ cells induces insulin resistance in mice. a** WT and Atg5 CD11c$^{KO}$ mice were fed HFD for 14 weeks according to the scheme, $n = 6$. **b** Total weight and **c** fasting blood glucose levels were measured every 2 weeks for 14 weeks. **d** Serum insulin concentrations were measured by ELISA after 24 weeks of HFD. Glucose tolerance test (**e**) and insulin tolerance test (**f**) were performed on WT and Atg5 CD11c$^{KO}$ mice fed HFD and treated for 14 weeks. The corresponding area under the curve was calculated for each group. **g** Photos of the dissected liver after perfusion. **h** Liver weights were measured after 14 weeks of HFD. **i** Serum (left) and hepatic (right) triglyceride levels was measured after 14 weeks of HFD. **j** Serum ALT was measured by colorimetric assay. **k** Representative hematoxylin and eosin-stained hepatic sections (×400), scale bars, 100 μm. **l** Histology score for steatosis, hepatocyte ballooning, lobular inflammation, NAFLD Activity Score, and fibrosis was quantified. Error bars are the mean ± SD, two-tailed Student's t test, $n = 5$ mice. Experiments were performed three times. Source data are provided as a Source Data file. Mouse image provided with permission from Servier Medical Art.

***Atg5* deletion in hepatic CD11c⁺ cells increases IL-23 production.** To investigate the mechanism of action associated with the induction of NAFLD development in CD11c⁺ cells from Atg5 CD11c$^{KO}$ mice, we analyzed the gene expression profile from control and Atg5 CD11c$^{KO}$ mice after being fed HFD for 14 weeks (Fig. 5a). Using RNA-sequencing (RNA-seq), we determined the effect of *Atg5* deletion in CD11c⁺ cells. The whole transcriptome is shown as a volcano plot based on the *p*-value (*p*-val) and expression FC of each analyzed gene (Fig. 5b). Genes that were significantly affected by *Atg5* deletion with a two-FC are either shown in orange (upregulated) or shown in blue (downregulated). We further processed our transcriptomic dataset with

ingenuity pathway analysis (IPA) and found that a network of genes that altogether statistically ($p = 3.59 \times 10^{-3}$) induced (*z*-score = 1472) liver inflammation in mice with the deletion of *Atg5* (Fig. 5c). The activation of this pathway is consistent with our previous observations that Atg5 CD11c$^{KO}$ mice were more prone to develop NAFLD compared to controls.

We have previously reported that CD11c⁺ cells from Atg5 CD11c$^{KO}$ mice had increased tissue inflammation, in particular among bone-marrow-derived DCs which exhibited increased production of IL-23[16]. Moreover, we observed that the IL-23 pathway was induced in the livers of patients with advanced NAFLD (Fig. 1a). Based on these observations, we depicted the

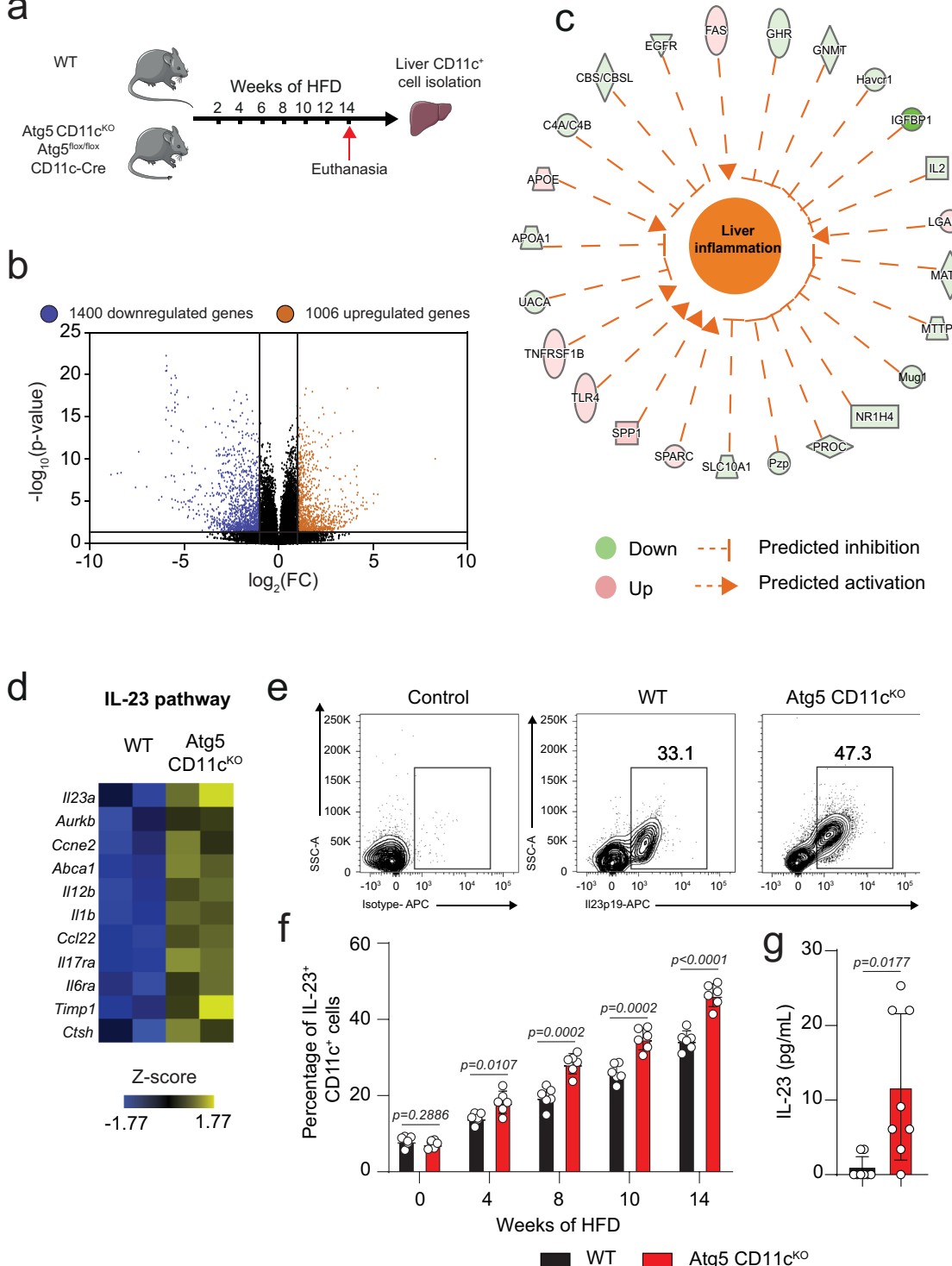

**Fig. 5 *Atg5* deletion in CD11c⁺ cells increases IL-23 induction. a** WT and Atg5 CD11c$^{KO}$ mice were fed HFD for 14 weeks according to the scheme, $n = 6$. After 14 weeks of HFD, hepatic CD11c⁺ cells were isolated. **b** Volcano plot comparison of whole transcriptome gene expression of sorted hepatic WT and Atg5 CD11c$^{KO}$ cells from mice fed HFD for 14 weeks, $n = 6$. Differentially expressed genes (*p*-value < 0.05), with changes of at least 2-fold change (FC) are shown in orange (upregulated) or blue (downregulated). **c** Network analysis of upregulated (red) and downregulated (green) genes overall significantly predicted to induce liver inflammation. **d** Heat plot of differentially expressed genes related to IL-23 pathway. **e** Representative flow cytometric plots of IL-23 expression in hepatic CD11c⁺ cells in WT and Atg5 CD11c$^{KO}$ mice after 14 weeks of HFD. **f** Percentage of hepatic IL-23⁺ CD11c⁺ cells after 0, 4, 8, 10, and 14 weeks of HFD, $n = 5$ mice. **g** Serum IL-23 levels in WT and Atg5 CD11c$^{KO}$ after 14 weeks of HFD, $n = 8$ mice. Error bars are the mean ± SD, two-tailed Student's *t* test. Experiments were performed three times. Source data are provided as a Source Data file. Mouse image provided with permission from Servier Medical Art.

expression of key genes associated with the IL-23 pathway in a heatmap (Fig. 5d). Genes having a critical function in the induction of the IL-23 process were induced, including *Ccl22, Il6ra*, and *Il-23a* coding for the IL-23p19 subunit in cells isolated from Atg5 CD11c[KO] mice. To confirm these results at the protein level, the expression of IL-23p19 in CD11c[+] cells isolated from Atg5 CD11c[KO] and control mice was examined using flow cytometry at different time points of HFD treatment (Fig. 5e, f). As seen in Fig. 5f, the percentage of IL-23[+]CD11c[+] cells increased the longer the mice were on HFD. IL-23 secretion is restricted to antigen-presenting cells such as CD11c[+] cells. As expected, the percentage of IL-23[+]CD11c[-]CD45[+] cells was low and not affected by *Atg5* deletion (Supplementary Fig. 4e). Although very low, we also measured IL-23 serum levels and confirmed that Atg5 CD11c[KO] mice have increased circulating concentrations of IL-23 compared to controls (Fig. 5g). We also assessed IL-23 expression in CD11c[+] of VAT from WT and Atg5 CD11c[KO] mice (Supplementary Figure 4f-g). The percentage of IL-23[+]CD11c[+] cells was significantly increased in VAT of Atg5 CD11c[KO] mice. Therefore loss of Atg5 in CD11c[+] cells induces IL-23 at multiple sites, as in addition to liver and VAT, it has previously been established that IL-23 expression is increased in the lungs of these mice[16].

Before HFD treatment (0 weeks), *Atg5* deletion did not affect IL-23p19 secretion in CD11c[+] cells, however, in response to the HFD treatment, we observed a significant increase in the percentage of IL-23p19[+]CD11c[+] cells in Atg5 CD11c[KO] mice compared to controls. Tang et al. have demonstrated that activation of the p38 pathway resulted in increased NAFLD development[22]. Moreover, another study by Yang et al. has also established that the activation of p38 was involved in IL-23 production by DCs[23]. In this context, we have measured by flow cytometry the activation of p38 by staining the active phosphorylated form of p38 in CD11c[+] cells isolated from Atg5 CD11c[KO] and control mice (Supplementary Fig. 4a). Notably, we observed that the active form of p38 was induced in CD11c[+] cells isolated from Atg5 CD11c[KO] mice, suggesting that *Atg5* deletion could activate the p38 mitogen-activated protein kinase (MAPK) leading to increased IL-23 production in CD11c[+] cells. To go further, immune cells isolated from Atg5 CD11c[KO] and WT mice fed HFD were stimulated ex-vivo in the presence of a specific p38-inhibitor or vehicle control. After treatment, the percentage of IL-23[+]CD11c[+] cells was quantified by flow cytometry. In response to the inhibitor treatment, Atg5 CD11c[KO] cells produced less IL-23 compared to WT cells, suggesting a dependence of IL-23 secretion on p38 signaling in Atg5 CD11c[KO] cells (Supplementary Fig. 4b). It has been suggested that p38 in association with NF-κB signaling could be involved in the IL-23 secretory pathway[24-26]. We measured by flow cytometry the expression of the active phosphorylated NF-κB subunit p65 and p52 in CD11c[+] cells isolated from Atg5 CD11c[KO] and control mice. The expression of the p65 subunit, part of the canonical NF-κB pathway was increased in Atg5 CD11c[KO] cells, whereas p52 expression which belongs to the non-canonical pathway was not affected (Supplementary Fig. 4c, d). These results suggest that the canonical NF-κB pathway and not the non-canonical pathway is activated in Atg5 CD11c[KO] cells, potentially leading to increased IL-23 production in CD11c[+] cells. Altogether, these results suggest that the deletion of *Atg5* in CD11c[+] cells activates IL-23 secretion likely in a p38/ NF-κB-dependent manner, inducing insulin resistance and NAFLD development.

**The preventive anti-IL-23 treatment protects against insulin resistance and NAFLD development.** Thus far, we observed that *Atg5* deletion in CD11c[+] cells induced increased insulin resistance and NAFLD development and is associated with increased IL-23 production. We hypothesized that inhibiting the IL-23 pathway could mitigate the development of these conditions. Atg5 CD11c[KO] and control mice were fed HFD for 14 weeks and were also treated with a blocking anti-IL-23 antibody (1 mg/week) or its isotype control starting at week 2 (Fig. 6a). Fasting blood glucose levels and weights were monitored after 14 weeks of HFD exposure (Fig. 6b, c). Similar to our previous observations, Atg5 CD11c[KO] mice had increased total body weight and fasting blood glucose levels when compared to control mice. However, the anti-IL-23 treatment prevented hyperglycemia in Atg5 CD11c[KO] mice ($p < 0.05$, Fig. 6c) while not affecting fasting blood glucose levels in WT mice. Anti-IL-23 treatment did not impact total body weights in any group. We also weighed and measured fasting blood glucose levels before starting the HFD and anti-IL-23 treatments in each cohort, and there was no difference (Supplementary Fig. 5a, b). After 14 weeks of treatment, increased insulin concentrations were observed in Atg5 CD11c[KO] mice, which was interestingly mitigated by treatment with the anti-IL-23 antibody (Fig. 6d). Glucose tolerance and insulin sensitivity were also tested and quantified after 14 weeks of treatment, and anti-IL-23 improved glucose tolerance and insulin sensitivity in Atg5 Cd11c[KO] mice (Fig. 6e, f). These results demonstrate that IL-23 blockade can prevent insulin resistance following exposure to HFD. After 14 weeks of HFD, we again weighed the livers and noticed that anti-IL-23 treatment prevented the development of the hepatomegaly previously observed in Atg5 CD11c[KO] mice (Fig. 6g). Similarly, serum ALT levels remained within normal limits in the group of Atg5 CD11c[KO] mice treated with IL-23 blocking antibody when compared to the isotype treatment (Fig. 6h). Importantly, on histological examination, we observed that anti-IL-23 treatment prevented the development of steatosis, ballooning, lobular inflammation, and reduced NAFLD scores in Atg5 CD11c[KO] mice when compared to isotype controls (Fig. 6i, j). IL-23 blockade also led to a significant decrease in the steatosis score in WT mice. We observed no effect of IL-23 blockade on VAT weight (Supplementary Fig. 5c). Neutralizing the IL-23 pathway prevented hepatic hypertriglyceridemia in both control and Atg5 CD11c[KO] mice (Fig. 6k). We also measured in the liver lysates the levels of IL-6 and IL-18 (Supplementary Fig. 5d, e) after 14 weeks of HFD. Interestingly, both cytokine levels are decreased in the Atg5 CD11c[KO] mice, demonstrating that anti-IL-23 treatment reduced the inflammation at the local level. We quantified by flow cytometry the percentage of CD11c[+] cells in the liver in response to anti-IL-23 treatment. IL-23 blockade repressed the induction of the CD11c[+] population in both control and Atg5 CD11c[KO] mice (Supplementary Fig. 5f, g). Overall, this data confirms that deletion of *Atg5* in CD11c[+] cells induces IL-23 activation, which in turn contributes to the development of insulin resistance and NAFLD. Conversely, blocking the IL-23 pathway can prevent insulin resistance, NAFLD development, and the recruitment of CD11c[+] cells.

**Anti-IL-23 ameliorates established insulin resistance and NAFLD.** Given our findings, we next investigated whether anti-IL-23 treatment could exert a therapeutic effect on mice with established insulin resistance and NAFLD. Atg5 CD11c[KO] and control mice were fed HFD for 8 weeks to establish insulin resistance and induce NAFLD. After 8 weeks of HFD, mice were either treated with the neutralizing anti-IL-23 antibody (1 mg/week) or isotype control for 6 weeks while on the HFD (Fig. 7a). After 8 weeks of HFD and before starting the anti-IL-23 treatment, we weighed the mice, measured fasting blood glucose levels, and quantified serum ALT levels (Supplementary

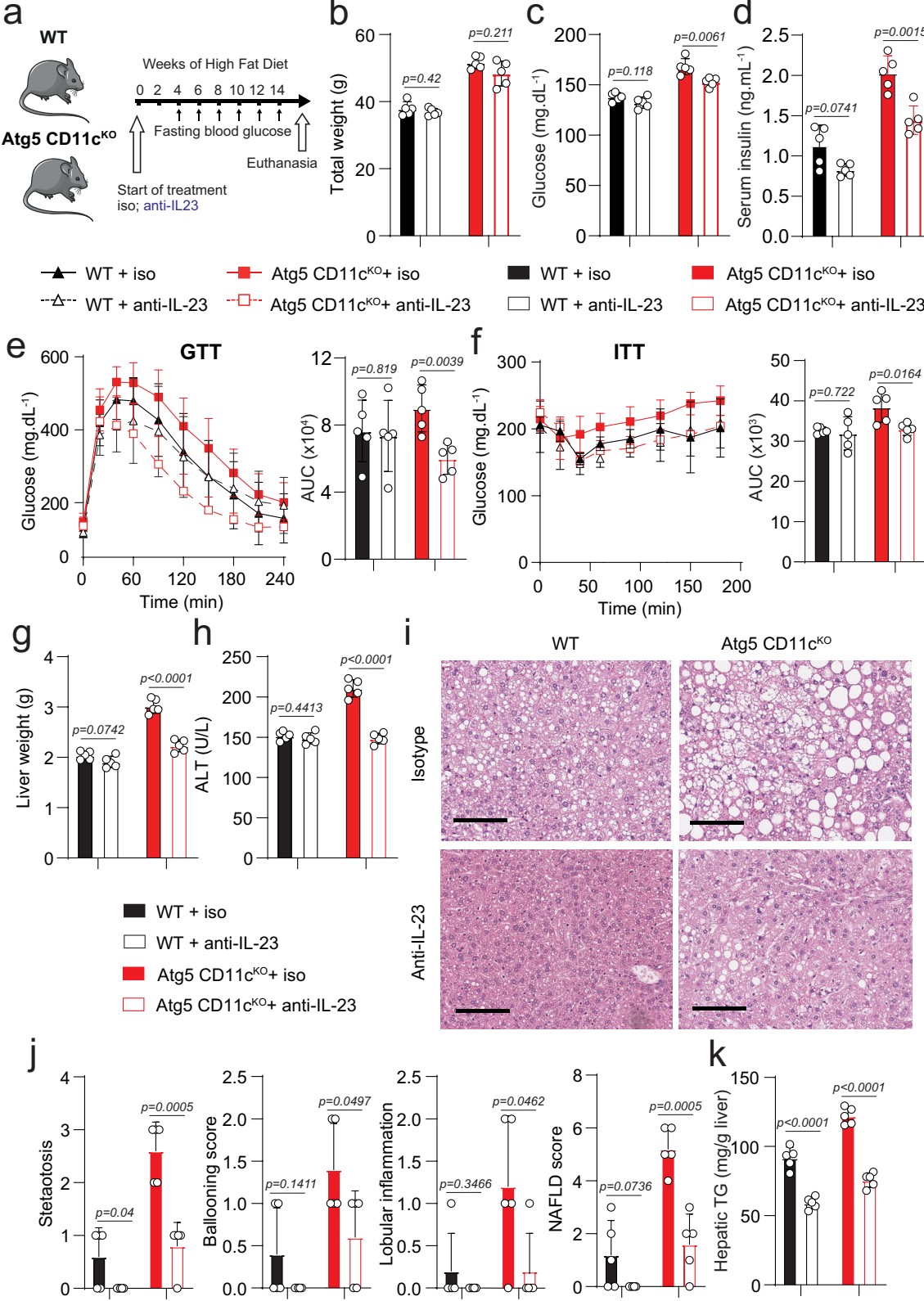

Fig. 6a–c). We confirmed that Atg5 CD11c^{KO} mice were heavier and had significantly higher glucose and ALT levels when compared to controls. There were no differences among mice of the same genotype before the beginning of the anti-IL-23 treatment. After 6 weeks of anti-IL-23 or isotype treatment, we weighed the mice and assessed fasting blood glucose and insulin levels (Fig. 7b–d). Similar to our previous observations,

anti-IL-23 treatment had an effect on glucose and insulin levels in Atg5 CD11c^{KO} mice, with no effect on WT mice. For both genotypes, total weights were not affected by anti-IL-23 treatment. Then glucose tolerance and insulin sensitivity were tested and quantified (Fig. 7e, f). Anti-IL-23 treatment improved glucose tolerance and insulin sensitivity in both WT and Atg5 CD11c^{KO} mice.

**Fig. 6 Preventive anti-IL-23 treatment protects against insulin resistance and NAFLD development in mice. a** WT and Atg5 CD11c<sup>KO</sup> mice fed HFD for 14 weeks were treated either with anti-IL-23 antibody or isotype control by intraperitoneal injection every week according to the scheme, n = 6. **b** Total weight, **c** blood glucose, and **d** serum insulin levels were measured after 14 weeks of diet. **e** Glucose tolerance test and **f** insulin tolerance tests were performed in WT and Atg5 CD11c<sup>KO</sup> mice fed HFD for 14 weeks and treated either with anti-IL-23 antibody or isotype control. The corresponding area under the curve was calculated for each group. **g** Liver weights were measured. **h** Serum ALT was measured by colorimetric assay. **i** Representative Hematoxylin and eosin-stained hepatic sections (×400), scale bars, 100 μm. **j** Histology score for steatosis, hepatocyte ballooning, lobular inflammation, fibrosis, and NAFLD Activity score was quantified. **k** Hepatic triglycerides levels were measured. Error bars are the mean ± SD, two-tailed Student's *t* test, n = 5 mice. Experiments were performed three times. Source data are provided as a Source Data file. Mouse image provided with permission from Servier Medical Art.

After 6 weeks of treatment of anti-IL-23 antibody or the isotype control, animals were euthanized, livers were weighed, and serum ALT levels were assessed (Fig. 7g, h). VAT weights were not affected by IL-23 neutralization (Supplementary Fig. 6d). The anti-IL-23 treatment prevented the development of hepatomegaly and prevented ALT elevation in Atg5 CD11c<sup>KO</sup> mice but had no effect in WT mice.

Histological analysis of livers from anti-IL-23 treated Atg5 CD11c<sup>KO</sup> mice also demonstrated an improvement of NAFLD scoring and decreased hepatic fat content (Fig. 7i–k). Anti-IL-23 treatment also improved NAFLD scoring and hepatic TG content in WT mice, while ALT levels remained within normal limits. These data indicate that IL-23 blockade has the potential to reverse established insulin resistance and NAFLD.

## Discussion
This study has identified a function for autophagy in the development of NAFLD and insulin resistance, via inducing IL-23 in CD11c<sup>+</sup> cells within the liver. We focused our exploration on the effect of *Atg5* deletion, a key protein in the formation of the phagophore vesicle specifically in CD11c<sup>+</sup> cells[27]. The rationale of this study was based on the observation that the expression of genes associated with autophagy was downregulated, whereas genes involved in the IL-23 pathway were upregulated in liver tissue from patients with advanced NAFLD. We also observed that the expression of the gene *ITGAX*, which codes the CD11c integrin, was induced in advanced NAFLD. In parallel, a previous study conducted by our laboratory has established that the IL-23 pathway was upregulated in DCs of mice deficient for Atg5[16]. Based on these observations, we designed experiments to assess the function of Atg5 on CD11c<sup>+</sup> cells in a cohort of mice on a high-fat diet and compared the results to normal chow control. Additionally, we characterized the metabolic parameters of the WT and Atg5 CD11c<sup>KO</sup> mice fed HFD. Our data suggest that the absence of autophagy in hepatic CD11c<sup>+</sup> cells results in increased weight gain, fasting hyperglycemia, insulin insensitivity, glucose intolerance, hepatomegaly, liver inflammation, and steatosis when compared to control mice. These changes are key contributors to the development of insulin resistance and the pathogenesis of NAFLD. Overall, the resulting inflammatory phenotype is similar to human NASH.

Consistent with human data, we have also demonstrated both at the mRNA and protein levels that the IL-23 pathway and IL-23 secretion were induced in Atg5 CD11c<sup>KO</sup> mice. Mechanistically, we believe this to be the result of activation of the NF-κB and p38 MAPK pathways as we observed that p38 and NF-κB p65 subunit were significantly more phospho-activated in hepatic myeloid cells from Atg5 CD11c<sup>KO</sup> mice. These findings are consistent with previous publications that have demonstrated important functions for p38 and NF-κB in NAFLD development[22,24–26,28,29]. Moreover, p38 activation has also been linked to increased IL-23 secretion in DCs[23]. Altogether these results suggest that autophagy deficiency in hepatic CD11c<sup>+</sup> cells induces the IL-23 pathway, and this induction is likely mediated

by p38 activation. By using a blocking antibody against IL-23, we were able to prevent the development of insulin resistance and NAFLD in Atg5 CD11c<sup>KO</sup> mice, suggesting that autophagy deletion causes IL-23 induction, resulting in metabolic derangements contributing to these distinct but related conditions. Most importantly, our data indicate that IL-23 blockade may be a relevant therapeutic pathway to reverse and ameliorate NAFLD. Our findings are clinically impactful as there are already several FDA-approved, commercially available, anti-IL-23 monoclonal antibody therapies such as Ustekinumab, Risankizumab, and Guselkumab for treatment of psoriasis and other inflammatory conditions, which could be repurposed for NASH/NAFLD[30–32].

Autophagy regulates the recycling of cytosolic nutrients as well as the degradation of dysfunctional organelles and consequently is a critical mechanism for the maintenance of liver homeostasis. Multiple liver pathologies have been associated with autophagy deficiency[9,33–35]. Previous studies have also established a strong connection between autophagy and cytokines production in myeloid cells[36]. Genetic knock-out mouse models of ATG core autophagy proteins have critical effects on their immune responses. Indeed the loss of the autophagy protein Atg16L1 in macrophages enhances IL-1β production in response to LPS[37]. Similarly, Atg7<sup>KO</sup> mice showed impaired pathogen clearance and the deletion of *Atg7* resulted in increased production of IL-1β[38]. Finally, the specific deletion of *Atg5* in macrophages has also been reported to induce cytokine secretion in myeloid cells in response to pathogens[39].

Interestingly, ATG proteins are involved in multiple mechanisms, Atg5 in particular is involved in innate antiviral immune responses, caspase-dependent apoptosis, adipogenesis and virus replication[40–46]. We recognize that there are some limitations to our work, including the fact that we cannot exclude the effect of Atg5 on other pathways beyond autophagy. However, it is important to point out that others have also shown an association between autophagy and the IL-23 pathway in other diseases[47,48]. For example, utilizing another autophagy-deficient model, it was demonstrated that Atg7 ablation in mononuclear cells increases IL-23 secretion in the context of Crohn's disease[47].

NAFLD is characterized by the aberrant accumulation of lipids in hepatocytes and the dysregulation of energy metabolism. Previous studies have shown that autophagy is negatively affected by the development of fatty liver disease and that activation of autophagy may resolve steatosis. Accumulating evidence has established that autophagy is dysfunctional during the progression and development of NAFLD. Indeed, impaired autophagy prevents the clearance of excessive lipid droplets, damaged mitochondria, and toxic protein aggregates, thus leading to the accumulation of intrahepatic fat (steatosis) and promoting liver inflammation. Deletion of *Atg5* in the liver induces hepatomegaly and liver injury and also resulted in increased apoptosis, inflammation, and fibrosis in the liver in mice[33]. Similar to autophagy deficiency in hepatocytes, our study demonstrates that *Atg5* deletion in hepatic CD11c<sup>+</sup> cells contributes to the development of hepatomegaly, steatosis, and progression to NAFLD.

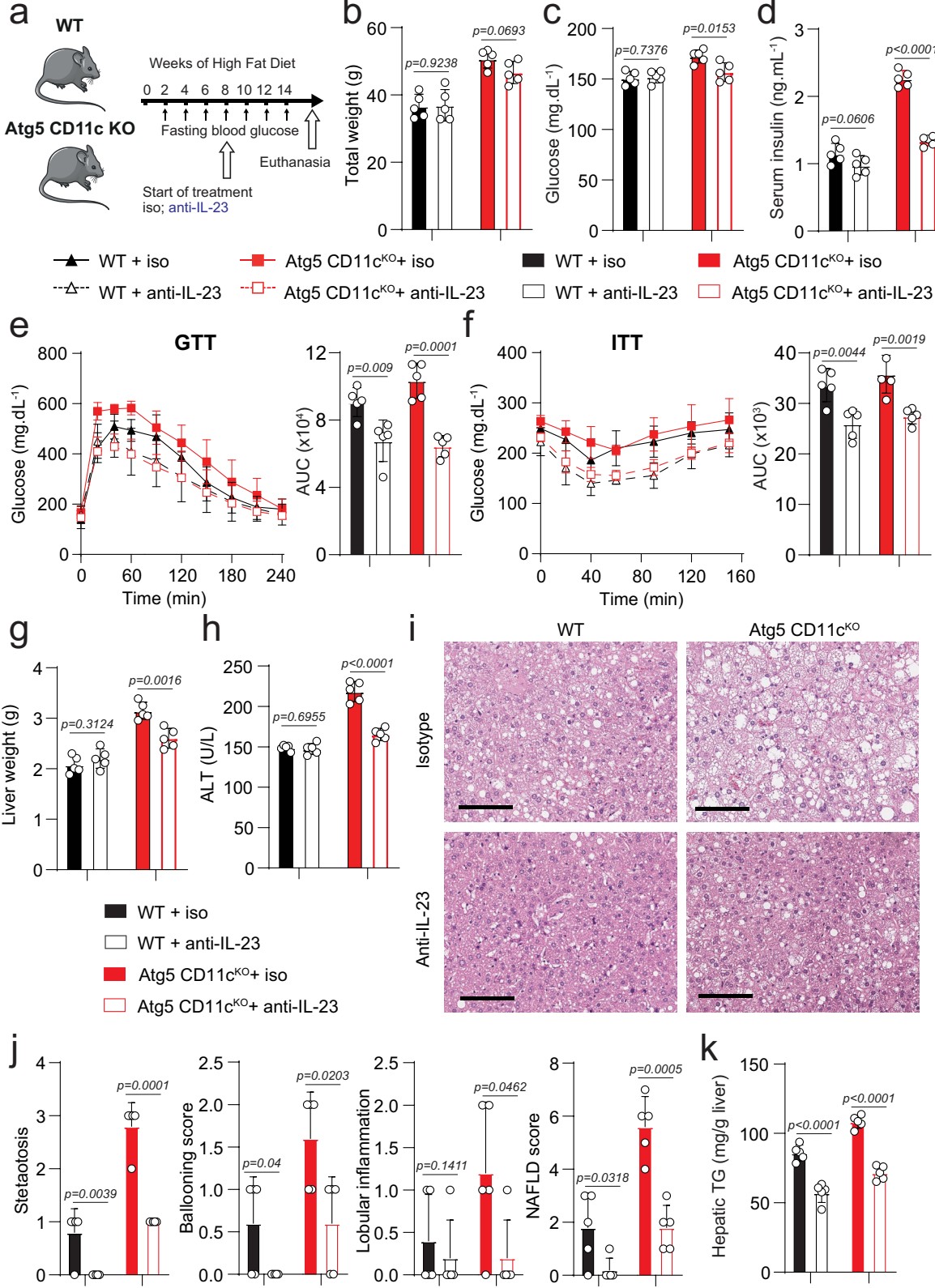

Homeostasis regulation and rapid response to tissue damage in the liver is mediated by crosstalk between the resident liver nonparenchymal cells (NPCs) and infiltrating inflammatory cells. Among liver NPCs, myeloid cells have an important activity in controlling local inflammation and promoting the regenerative function of the liver to maintain tissue homeostasis. Hepatic myeloid cells have a high heterogeneity and include resident KC,

monocyte-derived KCs (mo-KCs), and DCs[21]. Like most tissue-resident macrophages, KCs are derived from the differentiation of local precursors that originated from the yolk sac and colonized the liver during embryogenesis[49]. Steady-state KCs can be considered a homogeneous liver-resident macrophage population; however, in the context of inflammation, marrow-derived inflammatory macrophages can be recruited[50]. In the context of

**Fig. 7 Therapeutic anti-IL-23 treatment ameliorates established insulin resistance and NAFLD. a** WT and Atg5 CD11c[KO] mice-fed HFD for 14 weeks. After 8 weeks of diet, mice were treated either with anti-IL-23 antibody or isotype control by intraperitoneal injection every week according to the scheme, n = 6. **b** Total weight, **c** blood glucose, and **d** serum insulin levels were measured after 14 weeks of diet. **e** Glucose tolerance test and **f** insulin tolerance tests were performed in WT and Atg5 CD11c[KO] mice fed HFD for 14 weeks and treated either with anti-IL-23 antibody or isotype control. The corresponding area under the curve was calculated for each group. **g** Liver weights were measured. **h** Serum ALT was measured by colorimetric assay. **i** Representative Hematoxylin and eosin-stained hepatic sections (×400), scale bars, 100 μm. **j** Histology score for steatosis, hepatocyte ballooning, lobular inflammation, fibrosis, and NAFLD Activity score was quantified. **k** Hepatic triglycerides levels were measured. Error bars are the mean ± SD, two-tailed Student's t test, n = 5 mice. Experiments were performed three times. Source data are provided as a Source Data file. Mouse image provided with permission from Servier Medical Art.

obesity, chronic low-grade inflammation has been observed in metabolic tissues, notably due to increased circulating endotoxins and the development of leaky gut[51]. Prior studies have shown that HFD treatment can induce inflammation and is associated with the increased expression of CD11c in inflammatory myeloid cells[12–14]. These pro-inflammatory CD11c liver populations have also been reported to augment the cycle of inflammation in NASH in mice[52]. We characterized the expression of the CD11c levels in the different hepatic myeloid subpopulations and their respective frequencies. We noticed that the resident TIM4[+] KC which expressed lower levels of CD11c, had lower percentages in mice fed HFD whereas the pro-inflammatory recruited TIM4[-] KC expressed high levels of CD11c and their distribution increased, regardless of the mice genotype. Interestingly, a remarkable accumulation of DCs was observed in Atg5 CD11c[KO] mice under HFD conditions. It is worth mentioning here that the identification and characterization of the different hepatic macrophage populations have been the focus of multiple publications in recent years highlighting the critical activity of these subsets specifically in lipid-rich contexts and NASH[21,53,54].

In the present study, we observed that IL-23 blockade improves glucose tolerance, preserves insulin sensitivity, and supports liver function in Atg5 CD11c deficient mice, thus demonstrating that autophagy deficiency upregulates IL-23 secretion in CD11c[+] cells, contributing to NAFLD development. These data also establish that therapeutic strategies using anti-IL-23 treatment dampen the development of insulin resistance and NAFLD. Similarly, therapeutic anti-IL-23 treatment in Atg5 CD11c deficient mice proves that established insulin resistance and NAFLD can be reversed via this approach. WT mice treated with pre-emptive anti-IL-23 during exposure to HFD did not exhibit changes in insulin sensitivity or ALT levels but had reduced hepatic TG content. In the therapeutic groups, however, glucose tolerance, insulin sensitivity, and NAFLD score were improved in WT mice suggesting that the IL-23 pathway was more activated in this group. Based on this observation we hypothesize that because the anti-IL-23 treatment started after 8 weeks of HFD, WT mice had reached a level of inflammation and IL-23 activation sufficient to induce significant NAFLD damage. Unlike the WT mice from the therapeutic group, WT mice in the protective group did not develop a high level of chronic inflammation as the IL-23 pathway was blocked from the beginning of the HFD treatment. In support of our findings, several recent studies in mice and humans clearly suggest that *Il23* deletion among the CD11c population attenuates pathogenesis of inflammatory bowel disease[55,56]. Since IL-23 is produced by activated myeloid cells including macrophages and DCs and IL-23R is expressed by activated DCs and macrophages, IL-23 is able to drive an autocrine loop within the immune system, leading to the production of a number of inflammatory mediators that contribute to the pathologies associated with NASH and NAFLD[57].

In conclusion, the present study demonstrates that impaired autophagy in CD11c[+] cells causes metabolic dysregulations including NAFLD development and that these dysfunctions are mainly mediated by aberrant IL-23 induction. We have also determined that IL-23 blockade improves glucose tolerance and insulin sensitivity and pathologies associated with NAFLD not only in a preventive approach but also in a therapeutic approach. Clearly many questions remain to be answered before these results will be translated into better therapies for patients with NAFLD; however, our results suggest that the potential benefits make the effort worthwhile. Since anti-IL-23 is currently FDA approved for the treatment of other pro-inflammatory diseases, we believe our studies will open avenues for the development of promising strategies against the development and treatment of NAFLD.

## Methods

All experimentation protocols were approved by the USC Institutional Animal Care and Use Committee and conducted in accordance with the USC Department of Animal Resources' guidelines and the principles of the Declaration of Helsinki.

**Mice.** Atg5[flox/flox] and LC3-GFP mice, both on the C57BL/6J genetic background, are a gift from Dr. Noboru Mizushima (Tokyo Medical and Dental University, Tokyo, Japan). Mice were screened by using PCR, and Atg5[flox/flox] homozygote CD11c-Cre homozygote mice were selected for the experiments. Atg5[flox/flox] mice were backcrossed to B6.Cg-Tg(Itgax-cre)1-1Reiz/J mice (CD11c-Cre, #008068, Jackson Laboratory) and bred in our facility at the Keck School of Medicine under protocols approved by the Institutional Animal Care and Use Committee (IACUC). C57BL/6J mice were purchased from the Jackson laboratory. Four- to eight-week-old mice males were used in the studies. All mice were bred in our animal facility at the Keck School of Medicine, University of Southern California (USC) and maintained at a macroenvironmental temperature of 21–22 °C, humidity (48–52%), in a conventional 12:12 light/dark cycle with lights on at 6:00 a.m. and off at 6:00 p.m. For cohousing experiments mice, two WT control mice and two Atg5 CD11c[KO] mice were transferred to one common cage after weaning. WT control mice were ear-tagged to enable identification.

**Diet-induced obesity and in vivo treatments.** When indicated, mice were fed HFD ad libitum (Rodent diet with 60 kcal% Fat, cat#D12492i, Research Diets Inc., New Brunswick, New Jersey) for the indicated times. All other mice were fed NCD ad libitum (Cat#5053, TestDiet). For in vivo experiments investigating anti-IL-23 treatment, the neutralizing anti-mouse IL-23p19 (Janssen Pharmaceuticals) (1 mg/week) or the isotype control (Janssen Pharmaceuticals) was administered intraperitoneally every 7 days from the indicated start of treatment until termination of the experiment.

**In vivo metabolic phenotyping.** To measure weight and fasting blood glucose levels, mice were fasted overnight (~14–16 h), weighed and blood collected every 2 weeks. Glucose values were measured using a glucometer (Contour Next EZ, Bayer, Leverkusen, Germany). For intraperitoneal glucose tolerance tests (ip-GTT), mice were fasted overnight (~16 h), weighed, and injected with 2 g/kg 20% D-glucose (Sigma Aldrich) solution intraperitoneally. Blood glucose values were measured for each mouse by collecting a drop of blood before injection and at 20, 40, 60, 90, 120, 150, 180, 210, and 240 min post-injection. For insulin tolerance tests (ITT), mice were fasted for 5 h, weighed, and injected with 0.5U/kg human insulin (Novolin, Novo Nordisk, Bagsværd, Denmark) diluted in Sodium Chloride Solution 0.9% w/v (Azer Scientific, Morgantown, Pennsylvania) solution intraperitoneally. For both glucose and ITT, the data were analyzed by quantifying the AUC for each group of mice. When indicated, blood was collected by cardiac puncture and plasma insulin levels were measured using the ultrasensitive mouse insulin ELISA Kit (Cat#90080, Crystal Chem High-Performance Assays). The level of IL-23 was quantified using the LEGENDplex mouse inflammation panel (Cat#740151, BioLegend). Metabolic analysis of whole animals was performed using PhenoMaster/LabMaster home cages following the manufacturer's instructions (TSE Systems). Briefly, at the indicated time after the onset of treatment, mice

were singly housed, and measures were taken every 27 min for 5 days. Measures included oxygen consumption and carbon dioxide output, as variations in oxygen consumption and energy expenditure (heat) over time were calculated. Energy expenditure was normalized to body mass and lean mass. Mouse body composition was assessed with a minispec LF90 TD-NMR (Time-domain NMR spectroscopy) analyzer (Bruker) to quantify fat and lean mass[58].

**Liver preparation and flow cytometry**. Livers were collected at the indicated times after transcardial perfusion to clear organs of red blood cells. Livers samples were digested in collagenase IV (MP Biomedicals, LLC) at 37 °C for one hour and then processed on a 70 μm nylon cell strainer (Falcon) into a single cell suspension. Liver samples were resuspended in 30% Percoll solution and centrifuged at 750 g for 20 min without brake. Supernatants were discarded and pellets were resuspended in RBC Lysis Buffer (BioLegend) for 5 min. Cells were then washed and ready for staining. Stained cells were analyzed on FACSCanto II and/or FACSARIA III systems (Becton Dickinson). BD FACSDiva software v8.0.1 was used for flow cytometry data acquisition and analysis was performed using FlowJo version 10 software (TreeStar, Ashland, Oregon). The following mouse antibodies were purchased from eBioscience: FITC anti-mouse CD45 (dilution 1:200, clone 30-F11), eFluor 660 anti-mouse IL23p19 (1:200, fc23cpg), APC anti-mouse/human p-p38 MAPK (1:200, 4NIT4KK).

BV421 anti-mouse F4/80 (1:200, BM8), PE-Cy7 anti-mouse TIM4 (1:200, RMT4-54), APC-Cy7 or BV510 anti-mouse I-A/-E (1:200, M5/114.15.2), APC-eFluor780 or PerCP/Cy5.5 anti-mouse CD11c (1:200, N418), Alexa Fluor 647 anti-mouse CLEC4f (1:200, 3E3F9), BV650 anti-mouse CD45 (1:200, 30-F11), PE anti-mouse CD64 (1:200, X54-5/7.1), PE/Dazzle 594 anti-mouse/human CD11b (1:200, M1/70) and BV711 anti-mouse Ly6C (1:200, HK1.4) were purchased from BioLegend. Intranuclear staining was performed using the Foxp3 Transcription Factor Staining Kit (Cat#00-5523-00, ThermoFisher Scientific), according to the manufacturer's instructions. LIVE/DEAD fixable Aqua and Violet Dead Cell stain kits were purchased from (Dilution 1:500, Cat#L34966 & L34964, ThermoFisher Scientific). Intracellular staining was performed using the BD Cytofix/Cytoperm kit (Cat#554714, BD Bioscience, San Jose, CA), as previously described[59,60]. This kit contains Brefeldin A (named Golgiplug in the kit), thus allowing to quantify cytokine secretion by inhibiting vesicle trafficking. In some experiments, visceral adipose tissues (VAT) and spleens were processed for flow cytometry analysis as previously described[61,62].

**Metabolic colorimetric assays**. Serum alanine aminotransferase (ALT) levels were measured at the Analytical, Metabolomics and Instrumentation Core of the Research Center for Liver Disease at USC, according to manufacturer's instructions (Teco Diagnostics ALT/GPT). Serum and liver TG and cholesterol levels were measured according to manufacturer's instructions respectively with TG assay kit (Abcam—ab65336) and Cholesterol Assay kit (Abcam—ab65390).

**RNA sequencing (RNA-seq) and data analysis**. Livers were collected at the indicated times after transcardial perfusion to clear organs of red blood cells. Liver samples were digested in collagenase IV (MP Biomedicals, LLC) at 37 °C for one hour and then processed on a 70 μm nylon cell strainer (Falcon) into a single cell suspension. Liver samples were resuspended in 30% Percoll solution and centrifuged at 750g for 20 min without brake. Supernatants were discarded and pellets were resuspended in RBC Lysis Buffer (BioLegend) for 5 min. Cells were then washed and ready for magnetic separation. The CD11c MicroBead kit UltraPure (Cat#130-125-835, Miltenyi Biotec) was then used according to the manufacturer's conditions in order to isolate CD11c+ cells. Total RNA was isolated using MicroRNAeasy (Qiagen, Valencia, California) from freshly isolated hepatic CD11c+ cells. 10 ng of input RNA was used to produce cDNA for downstream library preparation. Samples were sequenced on a NextSeq 500 (Illumina) system as previously described[63–65]. Raw reads were aligned, normalized, and further analyzed using Partek Genomics Suite software, version 7.0; Partek Inc., St. Louis, MO, USA. Pathway analysis was performed using the Qiagen IPA software.

**Reverse transcription-polymerase chain reaction (RT-PCR) assays**. For measuring *Atg5* expression, mRNA was extracted and converted to cDNA for each mouse and quantified by Real Time-Polymerase Chain Reaction (RT-PCR). The following primers were used: *Atg5* (exon3-1): 5′-GAATATGAAGGCACACCCCT GAAATG-3′; *Atg5* (short2): 5′-GTACTGCATAATGGTTTAACTCTTGC-3′; *Atg5* (check2): 5′-ACAACGTCGAGCACAGCTGCGCAAGG-3′; *Atg5* (5L2): 5′-CAG GGAATGGTGTCTCCCAC-3′. Total RNA was extracted from isolated hepatic CD11c+ and CD11c− cells using the RNAasy mini kit (Cat#74004, Qiagen, Valencia, CA) and cDNA generated with the High-Capacity cDNA Reverse Transcription Kit (Cat#4368814, Applied Biosystems, Carlsbad, CA). RT-PCR was performed using CFX96 thermal cycler (Bio-Rad, Hercules, CA) and the ΔΔCt method was used for data analysis.

**Ex vivo treatment and cell culture**. Liver and VAT CD45+ cells from WT and Atg5 CD11cKO mice were cultured in RPMI (Gibco) supplemented with 10% heat-inactivated fetal bovine serum (FBS) and 100 units per mL penicillin–streptomycin,

with Golgiplug (BD Cytofix/Cytoperm kit, BD Bioscience, San Jose, CA) for 3 h. In some experiments, a selective inhibitor of p38 MAPK (SB 203580, TOCRIS) at 10 μM or vehicle (DMSO) was added to the culture. After treatment, cells were intracellularly stained with eFluor 660 anti-mouse IL23p19 (1:200, fc23cpg, Ebioscience). IL-23 expression in CD11c+ cells was evaluated by flow cytometry using FACSCanto II.

For immunoblotting, livers were processed and hepatic CD11c+ cells were sorted from C57BL/6 mice (NCD vs. HFD) using the FACSARIA III systems (Becton Dickinson) as described above. Cells were either lysed immediately after sorting or cultured in supplemented RPMI for 15 hours in the presence or absence of the autophagy inhibitor chloroquine (Sigma Aldrich) at 25 μM. Liver cells from LC3-GFP mice (NCD vs HFD mice) were cultured in the same conditions to assess LC3 accumulation in CD11c+ cells by flow cytometry.

**Western Blot analysis**. For immunoblotting, cells were lysed in RPMI lysis buffer (EMD Millipore) supplemented with protease inhibitor cocktail (Roche) for 30 min. Lysates were centrifuged (13,000×g at 4 °C for 15 min) and supernatants were collected. Protein concentrations were determined using the Pierce BCA Protein Assay Kit (Cat#23225, ThermoFisher Scientific), and samples were adjusted to the same concentration. The adjusted protein eluates were mixed with 6× Laemmli SDS sample buffer and heated at 95 °C for 5 min. Protein eluates were resolved by sodium dodecyl sulfate-polyacrylamide gel electrophoresis and transferred to a PVDF membrane (BioRad). Membranes were blocked with 5% BSA and probed with LC3B (1:250, Cell Signaling Technology), p62 (1:1000, Cell Signaling Technology), and β-actin (1:1000, Santa Cruz Biotechnology) at 4 °C overnight. Membranes were incubated with horseradish peroxidase-conjugated goat secondary antibodies (1:3000, Invitrogen). Immunodetection was achieved by ProSignal Pico Spray (Genesee Scientific) and detected on a ChemiDoc Imaging System (BioRad).

**Histologic analysis**. Liver histology was assessed using hematoxylin and eosin (H&E) staining on paraffin-embedded sections using standard commercially used methods. The presence of steatosis was confirmed using Oil-Red-O staining on frozen OCT-embedded sections using standard methods. The liver histology was evaluated by an expert liver pathologist who was blinded to the different treatments. The amount of steatosis (percentage of hepatocytes containing fat droplets) was scored as 0 (<5%), 1 (5–33%), 2 (>33–66%), and 3 (>66%). Hepatocyte ballooning was classified as 0 (none), 1 (few), or 2 (many cells/prominent ballooning). Foci of lobular inflammation were scored as 0 (no foci), 1 (<2 foci per 200× field), 2 (2–4 foci per 200× field), and 3 (>4 foci per 200× field). The NAS was computed from the grade of steatosis, inflammation, and ballooning. Stained sections were acquired using a Leica DME microscope and Leica ICC50HD camera (Leica, Wetzlar, Germany) and analyzed using Leica LAS EZ software.

**Human liver samples**. We analyzed data from a biorepository previously published[17]. This biorepository was approved by an Institutional Review Board and contains frozen liver biopsies and clinical data from NAFLD patients who underwent a diagnostic liver biopsy to grade and stage severity of disease as part of the standard of care. Two groups at the extremes of NAFLD formed the discovery cohort: "mild" NAFLD, defined as fibrosis stages 0 or 1 ($n = 40$) and thus, little probability of developing the clinically significant liver disease over the next one to two decades, and "severe" NAFLD, defined as fibrosis stage 3 or 4 ($n = 32$) and thus, the significant likelihood of developing liver-related morbidity and mortality over the same time. The groups were matched for gender, age (±5 years), and body mass index (BMI, kg/m$^2$) (±3 points).

**Mouse liver samples for RNA sequencing analysis**. We analyzed data from a biorepository previously published[20]. This biorepository was approved by an Institutional Review Board and all experiments were approved by the ACUC (Animal Care and Use Committee) of the NHLBI, NIH, and all methods were performed according to relevant guidelines and regulations from ACUC. Totally, 5 mg of liver tissue were homogenized using UltraThorax, and RNA was purified using TRIzol-RNA lysis reagent (ThermoFisher Scientific) according to manufacturer's instructions. Each sample represents one mouse fed either NCD or HFD (D12327, Research Diet, 40% calories from fat) for 7 weeks.

**Statistical analysis**. Experiments were repeated at least three times ($n = 4$–8 each) and data are shown as the representative of 3 independent experiments. A two-tailed student's t-test for unpaired data (95% confidence interval) was used for comparisons between each group using Prism Software analysis software version 8 (GraphPad Software Inc.). Error bars represent standard deviation. The degree of significance was indicated as: *$p < 0.05$, **$p < 0.01$, ***$p < 0.001$.

**Reporting summary**. Further information on research design is available in the Nature Research Reporting Summary linked to this article.

## Data availability

Source data are provided with this paper. The RNA-seq data generated in this study (WT vs Atg5 CD11cKO mice) have been deposited in the Genbank database under accession code GSE190819. Publicly available data were downloaded from the Gene Expression Omnibus database under accession codes GSE49541 (40 mild NAFLD versus 32 advanced NAFLD patients[17]; https://doi.org/10.1053/j.gastro.2013.07.047) and GSE88818 (WT mice fed NCD or HFD for seven weeks[20]; https://doi.org/10.1038/srep40220). Source data are provided with this paper.

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

## Acknowledgements

This article was financially supported by National Institutes of Health Public Health Service grants R01 ES025786, R01 ES021801, R01 HL144790, R01 HL151493, R01 AI145813, and R01 HL151769 (O.A.). We are grateful to USC Libraries Bioinformatics Service for assisting with data analysis, in particular Dr. Yong-Hwee E Loh, Meng Li, and Dr. Yibu Chen. The bioinformatics software and computing resources used in the analysis are funded by the USC Office of Research and the Norris Medical Library. We thank Dr. Chengyu Liang for providing help with data interpretation. We also thank the Analytical, Metabolomics and Instrumentation Core of USC Research Center for Liver Diseases, NIH P30 DK048522.

## Author contributions

L.G.-T. and D.G.H performed experiments, analyzed the data, designed the figures, and wrote the paper. C.Q. performed experiments and actively participated in the revision. E.H., B.P.H, and G.R.A.M. helped perform the experiments. P.S.J. and J.D.P contributed to data interpretation. A.I. and L.D. measured ALT levels and critically revised the paper. P.S., J.E., L.G.M. and H.R.R. provided study guide. O.A. supervised the studies, designed the experiments, conceptualized, interpreted the data, and finalized the paper.

## Competing interests

The authors declare no competing interests.
