## [Peer Review File · Nature Communications]

Autophagy impairment in liver CD11c+ cells promotes non-alcoholic fatty liver disease through production of IL-23REVIEWER COMMENTS

Reviewer #1 (Remarks to the Author):

The manuscript by Galle-Treger et al. seeks to understand the role of autophagy in CD11c expressing cells during the development of NALFD-NASH. To this end, they use a CD11c-Cre system to delete the autophagy protein ATG5. In response to a high fat diet the KO mice develop more severe obesity and NASH. The authors use RNA seq to suggest that increased IL-23 is a feature of the ATG5KO mice CD11c cells. Using an IL-23 neutralizing antibody they demonstrate that the insulin sensitivity and the NASH phenotype is diminished in the KO animals, suggesting that IL-23 may mediate this effect. The topic is of interest as NASH is a common, complex inflammatory condition for which additional mechanistic insight is needed. Overall the paper is well organized and the figures are clearly presented; however, there are several substantial deficiencies that detract from the impact of the current study. Please see my specific comments below.

Major:

1) In figure 2, the authors show that deletion of ATG5 from CD11c expressing cells increases obesity in response to high fat diet. Therefore, the observed liver changes are likely a consequence upstream changes in adiposity and circulating lipids. As this is an "obesity" phenotype rather than a liver specific phenotype, it is extremely unlikely that CD11c -expressing cells in the liver account for these observations. Do mice the CD11c-ATG5KO mice eat more food? Have a decreased metabolic rate? Have altered microbiota that changes nutrient absorption? Were WT and KO mice housed in the same cage or placed in separate cages based on genotype?

2) The assessment of CD11c expressing cells in the liver is inadequate. As noted, several subsets of myeloid cells and DCs can both express CD11c. The authors should have first used a panel of antibodies including F4/80, CD11b, Ly6C, MHCII, TIM4, and CLEC4F/VSIG4 to identify KCs, Mo-KCs, monocytes, MdMs (TIM4neg, F480 hi), cDC1s and cDC2 (Remmerie et al Immunity 2020 and Tran et al 2020). Assessing CD11c expression amongst these subsets in conjunction with addressing how these subsets change with ATG5 deletion would have more informative. A major limitation of the CD11c Cre model is that it will delete from both macrophages and DCs in multiple tissues. At a minimum, the investigators should also assess changes in macrophages and DCs in the adipose tissue as well as the liver.

3) The bulk sequence was done on total CD11c cells from the liver and therefore it is unclear whether differences in gene expression represent changes in cell composition or actual changes in gene expression (Figure 4). This significantly limits the interpretation of data.

4) The studies of autophagy and of the link to IL-23 are very superficial.

- The authors state that the LC3 GFP experiment presented in Figure 1C indicates impaired autophagy with HFD. However, this conclusion is not supported by the data. Static quantification of LC3 cannot be used to determine whether autophagic flux is increased or decreased. The authors would have to reassess LC3 levels in the presence of lysosome inhibitors. Moreover, based on their suggestions that autophagy may be diminished by HFD, it is harder to understand the rationale for further inhibiting autophagy with ATG5 KO
- The mechanistic link between decreased autophagy and IL-23 production is not addressed. The authors have previously shown that DCs with ATG5 KO release more IL-23, so this observation is not new. Why do IL-23 expressing cells appear to expand with ATG5KO?
- ATG proteins can also have roles outside of autophagy. Therefore, other genetic models of autophagy deficiency should be employed to determine the robustness of this association with IL-23.

5) The IL-23 inhibitory antibody data is little curious. The authors provide evidence that IL-23 producing CD11c cells increase significantly in WT and KO mice, with the KO mice possessing greater increase in these cells (Figure F). However, despite a large expansion in WT mice the neutralizing antibody had no effects on insulin resistance or metabolic parameters, save hepatic TAG. How do the authors explain this observation? Were IL-23 levels measured in the serum of WT and KO mice?

Minor

- 1) In Figure 7E and F the AUC curves have altered to Y axis to give the visual appearance of a more dramatic effect of the antibody. The y axis should go to zero.
- 2) Adipose tissue weights also need to be included the metabolic figures

Reviewer #2 (Remarks to the Author):

The authors found downregulation of autophagy factors and upregulation of IL-23 pathway factors and CD11c in a transcriptome database of the liver of NAFLD patients. As CD11c is expressed in myeloid cells under high-fat conditions, the authors created CD11c+ myeloid cell-specific ATG5KO mice. When fed a high-fat diet, these mice develop glucose intolerant, insulin resistance, and liver steatosis, and show increased production of IL-23 from myelocytes. Administration of anti-IL-23 antibodies to these mice concurrently with a high-fat diet suppresses the development of NALFD. Furthermore, the administration of anti-IL-23 antibodies even ameliorates the symptoms of already established NALFD.

The data are mostly convincing, and the manuscript is written clearly. However, there are a few concerns with this paper.

1. The autophagy data in Fig. 1 are not convincing. Since GFP-LC3 is degraded by autophagy, a decrease in the GFP-LC3 level could indicate activation, rather than suppression, of autophagy. The authors should carefully determine autophagic flux. For further details on evaluation of autophagy, please refer to recent guidelines (PMID: 26799652, 32851706, 32839099).

2. The liver phenotype of ATG5 CD11cKO mice is striking and surprising. It looks similar to that of hepatocyte-specific ATG KO mice. The authors should confirm that the expression of ATG5 is indeed suppressed specifically in CD11c+ myeloid cells but not in hepatocytes.

3. The authors mention increased IL-23 production from CD11c myelocytes in ATG5 CD11cKO mice. However, there seems no direct evidence that IL-23 is hyper-secreted. Is there any difference in serum IL-23 levels between control and ATG5 CD11cKO mice? Does IL-23 function only in the microenvironment?

4. There have been many papers describing the relationship between autophagy and cytokine production. For example, autophagy inhibition in macrophages stimulates production of inflammatory cytokines (reviewed in PMID: 30737475). These previous works should be appropriately discussed.

Reviewer #3 (Remarks to the Author):

In this study, Galle-Treger et al present very interesting data linking obesity, autophagy and non-alcoholic fatty liver disease. The authors observe that high fat diet (HFD) increased the percentage of CD11c+ cells in the liver with impaired autophagy, and that autophagy deletion using an Atg5fl/flxCD11c-Cre model worsened several disease parameters. Exploring the underlying mechanisms, the authors determined that liver Atg5fl/flxCD11c-Cre under HFD secreted increased levels of IL-23 and more importantly, that the preventive administration of anti-IL-23 neutralising antibodies ameliorated diseases parameters. Moreover, the authors showed that therapeutic administration of anti-IL-23 antibodies reduced disease parameters both in Atg5fl/flxCD11c-Cre and wt mice. The manuscript is well written and main conclusions are supported by experimental evidence. These results have a clear biological relevance and show a potential new use for a therapy (anti-IL-23) that is already approved for humans. Below I have listed some comments, questions and suggestions to improve the quality and relevance of the data.

Comments:

- Fig. 1a shows that autophagy-related molecules (including Atg5) are reduced in patients with advanced NAFLD in liver biopsies, together with increased expression of IL-23-related molecules. It will be of great interest to perform the same type of analysis (i.e., qPCR in livers for at least Atg5 and if possible, other markers showed in Fig. 1a) in wt mice under HFD. There may be publicly available microarray/RNAseq data that have already analysed this. These data can support how well does HFD reflect the NAFLD disease, increasing the biological and pre-clinical relevance of the data.

- Fig. 1b nicely shows the gating strategy for liver CD11c+ cells, I but miss some sort of quantification, i.e a graph representing % of liver CD11c+ cells in different mice in normal vs high fat diet, to add robustness to the data. The text states that "CD11c expression was strongly induced in hepatic immune cells, lines 253-254). Is that really induced CD11c+ expression in "resident" immune cells? Or is that CD11c+ cells are "recruited" to the liver? Can the authors clarify that in the main text?

- Regarding the Atg5fl/flxCD11c-Cre, I miss some control experiments. For example, under normal diet, are there any gross differences between wt and kos? I.e., are there normal numbers and/or percentages of CD11c+ cells in liver and spleen? Parameters such as liver size or glucose tolerance, are they normal in KO mice under normal diet?. These data are important to know if there are basal alterations, or if all defects are triggered only by HFD.

- The Cre systems are often leaky, I miss some control experiments using Atg5wt/wtxCD11c-Cre mice. For example, the number and/or percentage of liver CD11c+ cells, is it equally increased in HFD in CD11c-Cre+ vs wt? If authors have already performed any experiment using Atg5fl/wtxCD11c-Cre mice, it will be interesting to mention in the main text if heterozygous animals show any sort of defect.

- The experiments in Fig. 4e/f show IL-23 secretion in CD11c+ cells. Some experimental details are missing in legend/mat&meth. Were these experiments performed in presence of brefeldin A or monensin to prevent cytokine secretion to the media? Otherwise IL-23 secretion may be underestimated. In these experiments, it will be interesting to mention if IL-23 secretion is detected in other CD45+ cells apart from the CD11c+ population, or if only CD11c+ cells contain IL-23-secreting cells.

- Can IL-23 effects be systemic? I.e., are IL-23 levels increased in blood serum in wt vs ko under HFD?

- IL-23 effects, are they due to autocrine loop on IL-23R+ CD11c+ cells or effect is on another cells?. For example, liver CD11c cells+ can be isolated, stimulated in presence of exogenous IL-23 and measure p-p38/FACS or IL-23 mRNA by qPCR.

- Regarding anti-IL-23 treatment, does treatment change percentage and/or numbers of CD11c+ cells in the liver?

Minor comments

- Figure legends must state how many independent experiments have been performed, and if graphs are representative of one experiment or pooled data from different experiments. As it is now, one may conclude that experiments have been performed only once. This comment applies to all figures.

- The deviation of the data in all figures is reflected by the standard error of the mean (SEM). SEM is appropriate for technical replicates, but for biological replicates (as each mouse is), the standard deviation (sd) is more appropriate. This comment applies to all figures.

- In several graphs (2b-f, 3c,5e-f, 6b, 7e-f)s, the Y axis does not start in 0, which can be misleading. The reasons for not using a big and partially empty graph are understandable, but it

would be more appropriate to use graphs that start at 0, and then cut part of the scale (two segment graphs in Prism software).

- Supplemental figures 4c and 4c are not mentioned in the text.

We thank the editors and reviewers for their constructive feedback on our manuscript. In response to these comments, we performed additional experiments and adapted the manuscript accordingly. The new additions and clarifications strengthen our manuscript and we believe that our data clearly support the notion that autophagy among hepatic CD11c⁺ cells plays a significant role in NALFD. Please see our point-by-point response below.

REVIEWER COMMENTS

Reviewer #1 (Remarks to the Author):

The manuscript by Galle-Treger et al. seeks to understand the role of autophagy in CD11c expressing cells during the development of NALFD-NASH. To this end, they use a CD11c-Cre system to delete the autophagy protein ATG5. In response to a high fat diet the KO mice develop more severe obesity and NASH. The authors use RNA seq to suggest that increased IL-23 is a feature of the ATG5KO mice CD11c cells. Using an IL-23 neutralizing antibody they demonstrate that the insulin sensitivity and the NASH phenotype is diminished in the KO animals, suggesting that IL-23 may mediate this effect. The topic is of interest as NASH is a common, complex inflammatory condition for which additional mechanistic insight is needed. Overall the paper is well organized and the figures are clearly presented; however, there are several substantial deficiencies that detract from the impact of the current study. Please see my specific comments below.

Major:

1) In figure 2, the authors show that deletion of ATG5 from CD11c expressing cells increases obesity in response to high fat diet. Therefore, the observed liver changes are likely a consequence upstream changes in adiposity and circulating lipids. As this is an “obesity” phenotype rather than a liver specific phenotype, it is extremely unlikely that CD11c -expressing cells in the liver account for these observations. Do mice the CD11c-ATG5KO mice eat more food? Have a decreased metabolic rate? Have altered microbiota that changes nutrient absorption? Were WT and KO mice housed in the same cage or placed in separate cages based on genotype?

We understand the reviewer’s concerns. As the reviewer suggested, we have now performed additional experiments to address this point. We actively monitored food intake, energy expenditure and mouse activity by performing full metabolic analysis of animals. We observed that Atg5 CD11c^{KO} mice had increased food intake during the night, which was reflected with increased activity at night as eating is recorded as an activity. We did not observe any difference in energy expenditure normalized to total body weight suggesting that the metabolic rate of Atg5 CD11c^{KO} is not affected (**Supplementary Figures 3 d-f**).

Furthermore, for these studies mice were separated based on their genotype, however to address the reviewer’s concerns, we mixed and co-housed WT and Atg5 CD11c^{KO} mice in the same cages. After being fed a high fat diet (HFD) for 14 weeks, we measured fasting blood glucose, whole body and visceral adipose tissue weights as well as the liver total weight ratio. These results are now included in the manuscript as **Supplementary Figures 3g-j**. Briefly, co-hosting the WT and Atg5 CD11c^{KO} mice showed significant differences in phenotype and support our previous observations and that differences in phenotype are not dependent on the microbiota.

To address the reviewer’s comment about the “obesity” phenotype, we have performed additional experiments. We would like to highlight that Atg5 CD11c^{KO} mice fed with normal chow diet did not show liver inflammation. Only after these mice were fed with HFD, we observed increased liver inflammation and recruitment of proinflammatory CD11c⁺ cells to the liver (**CD11c⁺ TIM4⁺ CLEC4f⁺**

macrophage recruitment Figures 4c and 4d and histology Figure 5k). However, we agree with the reviewer that Atg5 CD11c^{KO} mice fed a HFD had both liver inflammation and systemic inflammation as evidenced by increased IL-23 serum levels (**Figure 6g**). Please note that IL-23 secretion is restricted to antigen presenting cells such as CD11c⁺ cells and since autophagy is depleted specifically in this population, our observations clearly suggest that CD11c are highly accountable for the phenotype we observe. Furthermore, when mice are treated with anti-IL-23, we observed a decrease in fasting blood glucose, an improvement in insulin sensitivity and a reduction in whole body weight (**Figure 7 and 8**). Moreover, we have also assessed the frequencies and levels of expression of CD11c in myeloid cells from the visceral adipose tissue (VAT) in NCD (**Supplementary Figures 1a-b**) and HFD (**Supplementary Figures 2a-b**). We observed no difference between these populations in WT and Atg5 CD11cKO mice. In conclusion, our phenotype is dependent on CD11c⁺ population and IL-23 in the liver.

2) The assessment of CD11c expressing cells in the liver is inadequate. As noted, several subsets of myeloid cells and DCs can both express CD11c. The authors should have first used a panel of antibodies including F4/80, CD11b, Ly6C, MHCII, TIM4, and CLEC4F/VSIG4 to identify KCs, Mo-KCs, monocytes, MdMs (TIM4^{neg}, F480^{hi}), cDC1s and cDC2 (Remmerie et al. Immunity 2020 and Tran et al 2020). Assessing CD11c expression amongst these subsets in conjunction with addressing how these subsets change with ATG5 deletion would have more informative. A major limitation of the CD11c Cre model is that it will delete from both macrophages and DCs in multiple tissues. At a minimum, the investigators should also assess changes in macrophages and DCs in the adipose tissue as well as the liver.

We thank the reviewer for this suggestion, and have now performed additional experiments (**Figures 3, 4 and Supplementary Figures 1-2**). We adapted a new gating strategy for the different myeloid cell subsets. As suggested by the reviewer we have added to our panel the recommended markers. We have also assessed the expression of the CD11c marker in the liver and visceral adipose tissue of the WT and Atg5 CD11c^{KO} mice to confirm the effect of CD11c Cre model in both NCD and HFD conditions. In mice fed NCD, we observed no difference in the frequencies in the macrophage and DC populations such as TIM4⁻ CLEC4f⁺ Kupffer cells (KC), TIM4⁺ CLEC4f⁺ KC, TIM4⁻ CLEC4f⁻ macrophages and F4/80-CD11c⁺ MHC II⁺ DCs (**Figures 3a-d**). However in the HFD, unlike NCD, the frequency of TIM4⁻ CLEC4f⁺ KCs was increased in Atg5 CD11cKO mice and these cells also expressed higher levels of CD11c (**Figure 4a-d**). Interestingly, it has been established (Remmerie et al. 2020) that TIM4⁻ CLEC4f⁺ KC were a monocyte-derived KC (moKCs) population which was increasingly recruited in HFD conditions. TIM4⁺ CLEC4f⁺ KC are described as resident KCs which tend to be replaced by moKCs in lipid-rich environment. We observed no difference on the other macrophage subpopulations and DC in WT and Atg5 CD11cKO mice (Figure 4d). As mentioned above, we have also assessed the frequencies and levels of expression of CD11c in myeloid cells from the visceral adipose tissue (VAT) in NCD (**Supplementary Figures 1a-b**) and HFD (**Supplementary Figures 2a-b**). We observed no difference between these populations in WT and Atg5 CD11cKO mice.

3) The bulk sequence was done on total CD11c cells from the liver and therefore it is unclear whether differences in gene expression represent changes in cell composition or actual changes in gene expression (Figure 6). This significantly limits the interpretation of data.

We agree with the reviewer that the perfect approach would be to perform a single cell RNAseq which we are considering for future studies and we believe it is out of the scope of this study. While these exploratory RNAseq results were helpful, we would like to emphasize that we confirmed the critical findings at the at the protein level. Furthermore, since IL-23 secretion is restricted to antigen

presenting cells, the RNAseq of hepatic CD11c⁺ population was enormously helpful to delineate the IL-23 pathway in these cells (**Figure 6d**). Based on our RNA-seq analysis we have concluded that the IL-23 pathway is activated in Atg5 CD11cKO mice, and this result is confirmed by flow cytometry (**Figures 6e-f**).

4) The authors state that the LC3 GFP experiment presented in Figure 1C indicates impaired autophagy with HFD. However, this conclusion is not supported by the data. Static quantification of LC3 cannot be used to determine whether autophagic flux is increased or decreased. The authors would have to reassess LC3 levels in the presence of lysosome inhibitors. Moreover, based on their suggestions that autophagy may be diminished by HFD, it is harder to understand the rationale for further inhibiting autophagy with ATG5 KO

To further establish our model, we have assessed by Western Blots the LC3-II/LC3-I ratio and quantified p62 expression (**Figures 2c-d**). The conversion of LC3-I into LC3-II by a complex of ATG proteins has been well-established as an indicator of autophagy flux. In **Figures 2c-d**, we have shown that the LC3-II/LC3-I ratio was decreased in CD11c⁺ Atg5 CD11cKO cells. The p62 protein interacts with autophagic substrates and delivers them to autophagosomes for degradation. In the process, p62 is itself degraded and when autophagy is induced, a corresponding decrease in p62 levels is observed. We observed that p62 levels were increased in CD11c⁺ Atg5 CD11c KO cells compared to controls (**Figures 2c-d**). Therefore, the new results of Western Blot support our original observation in LC3-GFP mice.

5) The mechanistic link between decreased autophagy and IL-23 production is not addressed. The authors have previously shown that DCs with ATG5 KO release more IL-23, so this observation is not new. Why do IL-23 expressing cells appear to expand with ATG5KO?

Our understanding of the mechanisms regulating IL-23 secretion is currently very limited. However, our data suggests the involvement of p38 kinase in the CD11c⁺ lacking autophagy (**Supplementary Figure 4a**). To clarify the link between this observation and IL-23 secretion, we have conducted additional experiments using a p38 inhibitor. Briefly, hepatic cells isolated from WT and Atg5 CD11cKO mice were stimulated ex-vivo and treated with p38 inhibitor. In response to this inhibition, we observed reduced secretion of IL-23 by CD11c⁺ cells in Atg5 CD11cKO mice (**Supplementary Figure 4b**). We need to point out that we also identified that the expression of the active phosphorylated p65 subunit of the canonical NF- κ B pathway was increased in Atg5 CD11cKO mice (**Supplementary Figure 4c**). Therefore, the lack of Atg5 in CD11c⁺ cells promotes inflammatory pathways such as NF- κ B p65 and p38 resulting in increased inflammation status within hepatic CD11c⁺ cells.

6) ATG proteins can also have roles outside of autophagy. Therefore, other genetic models of autophagy deficiency should be employed to determine the robustness of this association with IL-23.

We agree with the reviewer that ATG5 may have some bystander effect and it is helpful to assess the level of IL-23 in other Autophagy models. Interestingly, several recent reports in the context of IBD and gut inflammation reported that lack ATG16L is associated with increased levels of IL-23 in gut antigen presenting cells and macrophages (PMID 30765845, 32474165). It has also been shown that Atg7 ablation in mononuclear cells increases IL-23 secretion in the context of Crohn's disease. Nevertheless, we cited those references and discussed limitation of our studies in the discussion.

7) The IL-23 inhibitory antibody data is little curious. The authors provide evidence that IL-23 producing CD11c cells increase significantly in WT and KO mice, with the KO mice possessing greater

increase in these cells (Figure F). However, despite a large expansion in WT mice the neutralizing antibody had no effects on insulin resistance or metabolic parameters, save hepatic TAG. How do the authors explain this observation? Were IL-23 levels measured in the serum of WT and KO mice?

As the reviewer mentioned, the IL-23 treatment had indeed a strong effect on glucose homeostasis and liver inflammation in Atg5 CD11cKO mice but had little effect in WT mice. Our hypothesis is that the HFD did not induce a level of inflammation high enough to significantly impair these metabolic parameters. WT mice fed a HFD have not reached a threshold of chronic inflammation where IL-23 neutralization can have an effect on these glucose homeostasis and liver inflammation. In fact, even after 14 weeks of HFD, WT mice weighed around 38g whereas Atg5 CD11cKO mice weighed 50g (Figure 5b). Notably, metabolic parameters were decreased in Atg5 CD11cKO mice in response to anti-IL-23 treatment to levels similar to those observed in WT mice (Figures 7d-k).

To address the reviewer's comment, we have also added the IL-23 serum levels measured in WT and Atg5 CD11cKO mice after 14 weeks of HFD (Figure 6g), consistently with our flow cytometry results, IL-23 levels were significantly higher in Atg5 CD11cKO mice.

Minor

1) In Figure 7E and F the AUC curves have altered to Y axis to give the visual appearance of a more dramatic effect of the antibody. The y axis should go to zero.

We have adapted the figures based on the reviewer's suggestions (Figures 7e-f).

2) Adipose tissue weights also need to be included the metabolic figures

Visceral adipose tissue weights were added to the corresponding figures (Supplementary Figures 3b, 5c and 6d).

Reviewer #2 (Remarks to the Author):

The authors found downregulation of autophagy factors and upregulation of IL-23 pathway factors and CD11c in a transcriptome database of the liver of NAFLD patients. As CD11c is expressed in myeloid cells under high-fat conditions, the authors created CD11c+ myeloid cell-specific ATG5KO mice. When fed a high-fat diet, these mice develop glucose intolerant, insulin resistance, and liver steatosis, and show increased production of IL-23 from myelocytes. Administration of anti-IL-23 antibodies to these mice concurrently with a high-fat diet suppresses the development of NALFD. Furthermore, the administration of anti-IL-23 antibodies even ameliorates the symptoms of already established NALFD.

The data are mostly convincing, and the manuscript is written clearly. However, there are a few concerns with this paper.

1) *The autophagy data in Fig. 1 are not convincing. Since GFP-LC3 is degraded by autophagy, a decrease in the GFP-LC3 level could indicate activation, rather than suppression, of autophagy. The authors should carefully determine autophagic flux. For further details on evaluation of autophagy, please refer to recent guidelines (PMID: 26799652, 32851706, 32839099).*

As suggested by the reviewer, we assessed by Western Blots the LC3-II/LC3-I ratio and p62 expression (Figure 2c and 2d). The conversion of LC3-I into LC3-II by a complex of ATG proteins has been well-established as an indicator of autophagy flux. We observed that the LC3-II/LC3-I ratio was decreased

in CD11c⁺ Atg5CD11c^{KO} cells (**Figures 2c-d**). The p62 protein interacts with autophagic substrates and delivers them to autophagosomes for degradation. In the process, p62 is itself degraded and when autophagy is induced, a corresponding decrease in p62 levels is observed. We observed that p62 levels were increased in CD11c⁺ Atg5 CD11c KO cells compared to controls (**Figures 2c-d**). On the contrary, when autophagy is potently inhibited, i.e. genetic knock out of ATG core autophagy proteins like ATG5, p62 accumulates in the cell.

2) The liver phenotype of ATG5 CD11cKO mice is striking and surprising. It looks similar to that of hepatocyte-specific ATG KO mice. The authors should confirm that the expression of ATG5 is indeed suppressed specifically in CD11c⁺ myeloid cells but not in hepatocytes.

As recommended by the reviewer we have measured the relative expression of Atg5 in CD11c⁺ and CD11c⁻ liver cells by RT-qPCR in both WT and ATG5 CD11cKO mice (**Supplementary Figure 1e**). Our results confirmed that Atg5 expression was specifically deleted in CD11c⁺ cells in ATG5 CD11cKO cells. There was no difference observed on Atg5 expression in CD11c⁻ cells between WT and Atg5CD11cKO cells.

3) The authors mention increased IL-23 production from CD11c myelocytes in ATG5 CD11cKO mice. However, there seems no direct evidence that IL-23 is hyper-secreted. Is there any difference in serum IL-23 levels between control and ATG5 CD11cKO mice? Does IL-23 function only in the microenvironment?

We understand the reviewer's concern and we have measured the IL-23 serum levels in WT and Atg5 CD11cKO mice after 14 weeks of HFD (**Figure 6g**). Consistently with our previous observation of increased IL-23 expression in CD11c⁺ cells, we observed that IL-23 serum concentrations were significantly higher in Atg5 CD11cKO mice after 14 weeks of HFD compared to WT mice.

The IL-23 receptor is specifically expressed by lymphoid cells expressing the ROR γ t transcription factor (PMID 25145755) suggesting that in association with low circulating levels of IL-23 that the action of IL-23 in the systemic environment would be quite limited and favor an effect on the microenvironment.

4) There have been many papers describing the relationship between autophagy and cytokine production. For example, autophagy inhibition in macrophages stimulates production of inflammatory cytokines (reviewed in PMID: 30737475). These previous works should be appropriately discussed.

We agree with the reviewer's recommendations have developed our discussion about the effect of autophagy on cytokine production and have cited the recommended publications in the discussion (PMID: 30737475, 18849965, 28258192, 30297526).

Reviewer #3 (Remarks to the Author):

In this study, Galle-Treger et al present very interesting data linking obesity, autophagy and non-alcoholic fatty liver disease. The authors observe that high fat diet (HFD) increased the percentage of CD11c⁺ cells in the liver with impaired autophagy, and that autophagy deletion using an Atg5fl/flxCD11c-Cre model worsened several disease parameters. Exploring the underlying mechanisms, the authors determined that liver Atg5fl/flxCD11c-Cre under HFD secreted increased levels of IL-23 and more importantly, that the preventive administration of anti-IL-23 neutralizing

antibodies ameliorated disease parameters. Moreover, the authors showed that therapeutic administration of anti-IL-23 antibodies reduced disease parameters both in Atg5^{fl/fl}CD11c-Cre and wt mice. The manuscript is well written and main conclusions are supported by experimental evidence. These results have a clear biological relevance and show a potential new use for a therapy (anti-IL-23) that is already approved for humans. Below I have listed some comments, questions and suggestions to improve the quality and relevance of the data.

Comments:

1) Fig. 1a shows that autophagy-related molecules (including Atg5) are reduced in patients with advanced NAFLD in liver biopsies, together with increased expression of IL-23-related molecules. It will be of great interest to perform the same type of analysis (i.e., qPCR in livers for at least Atg5 and if possible, other markers showed in Fig. 1a) in wt mice under HFD. There may be publicly available microarray/RNAseq data that have already analysed this. These data can support how well does HFD reflect the NAFLD disease, increasing the biological and pre-clinical relevance of the data.

We have analyzed a WT mouse RNA-seq study (PMID 28071704) providing the transcriptional profile of total livers (NCD vs HFD). Similarly to our results from **Figure 1a** in human liver samples, we have observed that autophagy pathway was significantly repressed after HFD whereas the IL-23 associated genes were induced (**Figure 1b**).

2) Fig. 1b nicely shows the gating strategy for liver CD11c⁺ cells, I miss some sort of quantification, i.e a graph representing % of liver CD11c⁺ cells in different mice in normal vs high fat diet, to add robustness to the data. The text states that "CD11c expression was strongly induced in hepatic immune cells, lines 253-254). Is that really induced CD11c⁺ expression in "resident" immune cells? Or is that CD11c⁺ cells are "recruited" to the liver? Can the authors clarify that in the main text?

We have added the corresponding quantifications in **Figure 2a**. To further our characterization of the different myeloid populations in the liver and their respective expression of the CD11c marker, we have added new markers to our panel such Ly6c, CD64, CD11b, TIM4 and Clec4f. These markers allowed us to differentiate the "recruited" inflammatory immune cells from the "resident" immune cells as explained in our Result section describing the updated **Figures 2 and 3**. We observed no difference in the macrophage subpopulations in WT and Atg5^{CD11c} KO mice fed NCD (**Figures 3a-d**), however we observed a specific increase of the frequency of TIM4⁻ CLEC4f⁺ Kupffer cells (KCs) in Atg5^{CD11c} KO mice in HFD (**Figures 4a-d**). Our observations are in line with previous reports by Remmerie et al. suggesting that TIM4⁻ CLEC4f⁺ KCs are monocyte-derived KCs recruited upon HFD exposure, replacing the resident KC identified as TIM4⁺ CLEC4f⁺ KCs. Strikingly, we also observed that these moKCs expressed higher levels of CD11c in Atg5^{CD11c} KO mice compared to controls (**Figures 4b-c**).

3) Regarding the Atg5^{fl/fl}CD11c-Cre, I miss some control experiments. For example, under normal diet, are there any gross differences between wt and kos? I.e., are there normal numbers and/or percentages of CD11c⁺ cells in liver and spleen? Parameters such as liver size or glucose tolerance, are they normal in KO mice under normal diet?. These data are important to know if there are basal alterations, or if all defects are triggered only by HFD.

We have observed no difference in the phenotypes of WT and Atg5^{CD11c} KO mice on NCD. We have added in **Supplementary Figures 3k-n**, total weights, liver weights, fasting blood glucose levels and ALT

levels. We have also quantified the different macrophage subpopulations and DCs in mice fed with NCD (**Figures 3a-d**) and observed no difference between WT and Atg5 CD11cKO mice on NCD. We have added the percentage of CD11c+ cells in the spleen (**Supplementary Figures 1c-d**) and observed no difference between WT and Atg5 CD11cKO mice on NCD. Altogether, these results establish that there is no difference in the phenotype of Atg5CD11cKO mice compared to controls when fed with NCD.

4) The Cre systems are often leaky, I miss some control experiments using Atg5wt/wtxCD11c-Cre mice. For example, the number and/or percentage of liver CD11c+ cells, is it equally increased in HFD in CD11c-Cre+ vs wt? If authors have already performed any experiment using Atg5fl/wtxCD11c-Cre mice, it will be interesting to mention in the main text if heterozygous animals show any sort of defect.

We understand the reviewer's concerns, however the CD11c-Cre mouse model is a well-established mouse model. Notably the specificity of this mouse model has been published in 2010 by Lee et al. in *Immunity* (PMID 20171125). In the Supplementary Figure 2, the authors analyzed splenocytes, lymph node cells and bone marrow cells from CD11c-Cre-GFP Tg mice. They showed that only CD11c expressing cells were GFP positive.

5) The experiments in Fig. 4e/f show IL-23 secretion in CD11c+ cells. Some experimental details are missing in legend/mat&meth. Were these experiments performed in presence of brefeldin A or monensin to prevent cytokine secretion to the media? Otherwise IL-23 secretion may be underestimated. In these experiments, it will be interesting to mention if IL-23 secretion is detected in other CD45+ cells apart from the CD11c+ population, or if only CD11c+ cells contain IL-23-secreting cells.

We now added in our **Methods section** the details of our protocol. We used the BD Cytofix/Cytoperm kit (BD Bioscience, San Jose, CA) containing Brefeldin A (labelled Golgi plug in the kit) in our ex-vivo stimulation experiments. As reported before, IL-23 expression is restricted to myeloid antigen presenting cells and our results also indicate that IL-23 in CD11c- cells is near background in WT and Atg5 CD11c KO mice (**Supplementary Figure 4e**).

6) Can IL-23 effects be systemic? I.e., are IL-23 levels increased in blood serum in wt vs ko under HFD?

We understand the reviewer's concern and we have added the IL-23 serum levels measured in WT and Atg5 CD11cKO mice after 14 weeks of HFD (**Figure 6g**). We observed that IL-23 concentrations were significantly higher in Atg5 CD11c KO mice after 14 weeks of HFD compared to WT mice. This result is consistent with the increase of IL-23 expression in CD11c+ cells measured by flow cytometry in Atg5 CD11cKO mice after HFD.

7) IL-23 effects, are they due to autocrine loop on IL-23R+ CD11c+ cells or effect is on another cells?. For example, liver CD11c cells+ can be isolated, stimulated in presence of exogenous IL-23 and measure p-p38/FACS or IL-23 mRNA by qPCR.

The receptor of IL-23 is specifically expressed by lymphoid cells expressing ROR γ t transcription factor (PMID 25145755), myeloid cells such as CD11c+ cells would not express this receptor and would not be able to respond to IL-23 stimulation.

8) Regarding anti-IL-23 treatment, does treatment change percentage and/or numbers of CD11c+ cells in the liver?

In response to IL-23 treatment the percentages of CD11c+ cells were significantly reduced in both WT and Atg5 CD11c^{KO} mice after anti-IL-23 treatment (**Supplementary Figure 5f-g**) confirming that IL-23 neutralization reduces inflammation in the liver.

Minor comments

- *Figure legends must state how many independent experiments have been performed, and if graphs are representative of one experiment or pooled data from different experiments. As it is now, one may conclude that experiments have been performed only once. This comment applies to all figures.*

We have added in the manuscript the number of times each experiment has been performed.

- *The deviation of the data in all figures is reflected by the standard error of the mean (SEM). SEM is appropriate for technical replicates, but for biological replicates (as each mouse is), the standard deviation (sd) is more appropriate. This comment applies to all figures.*

We have updated our figures according to the reviewer's recommendations.

- *In several graphs (2b-f, 3c, 5e-f, 6b, 7e-f), the Y axis does not start in 0, which can be misleading. The reasons for not using a big and partially empty graph are understandable, but it would be more appropriate to use graphs that start at 0, and then cut part of the scale (two segment graphs in Prism software).*

We have corrected our figures according to the reviewer's recommendations.

- *Supplemental figures 4c and 4c are not mentioned in the text.*

We have addressed this issue and mentioned these Supporting **Figures 5d-e** in the manuscript.

REVIEWER COMMENTS

Reviewer #1 (Remarks to the Author):

The revised manuscript by Galle-Treger et al is improved and the authors have made a good effort to address the concerns of this reviewer. However, there are still a few issues that should be addressed:

1) The authors were very responsive to my critique that the primary phenotype of the CD11c-ATG5KO is a difference in obesity. Based on their studies, it appears that the KO mice are hyperphagic and eat more food at nighttime leading excessive weight gain. These findings suggest that IL-23 may be acting centrally to regulate satiety centers. I am still not convinced that CD11c cells in the liver are responsible for this phenotype. The authors claim that it can't be related to adipose tissue macrophages because there is no difference in the % of CD11c expressing macrophages between the genotypes. However, the total number of CD11c expressing ATMs is actually much greater in the KO because the perigonadal fat pads are nearly double in size in the KO (and therefore for the same percentage of cells the absolute number of these macrophages per fat pad is significantly increased). Moreover, the amount of IL-23 production in the ATMs from WT and KO mice is not shown. Again, as CD11c-cre deletes throughout the body it cannot be concluded that the hepatic cells are responsible for this phenotype (this is related to point 3 below). How do the authors exclude that increased IL-23 from DCs is not the reason for the phenotype?

2) The authors did additional flow cytometry to better quantify the changes in specific macrophage populations with obesity. I am concerned, however, about some of their CLEC4F staining. From several recent papers (PMID: 32888418, 32562600, 33440159), we now know that KCs are depleted and TIM4neg monocyte derived cells enter the liver during NASH development. The TIM4neg cells can be subdivided into Mo-KCs (CLEC4Fpos, VSIG4pos) and "Lipid associated macrophages" referred to as LAMs. The LAMs can also be further subdivided into C-LAMs and LAMs based on the expression of CCR2/Cx3cr1 (PMID:33440159). In general, about 50% of the TIMneg cells are Mo-KCs and the other 50% are LAM subsets. However, in Figure 4 of this manuscript nearly all (96%) of the TIM4neg cells appear to express CLEC4F. This would be extremely surprising. Did the authors also use an isotype control antibody? I am concerned that some of this staining may be non-specific. Based on our own experience, CLEC4F staining by flow can be challenging. This is relevant because it has been shown that C-LAMs and LAMs have increased CD11c expression and that KCs and Mo-KCs do not (PMID: 32888418, 33440159). Thus, the figure legend title for Fig 4 is misleading and should be changed. In addition, the discussion paragraph about hepatic myeloid cells needs to address this point and should more thoroughly incorporate the findings for the 3 manuscripts referenced above.

3) As a more minor point, the box labeled monocytes (Fig 3A, Fig 4A) actually contains both neutrophils and monocytes (monocytes are higher in Ly6C and lower in CD11b, whereas the neutrophils are intermediate Ly6C and higher in CD11b. In addition, the neutrophils are higher SSC. Using these parameters, the authors could re-gate their data to remove the neutrophils.

4) I am still a little unclear about how the authors integrate their data. Since the mice eat more food when fed a HFD they get more obese which leads to more hepatic steatosis and liver injury. This will then cause more LAM infiltration and more KC loss (in our experience the bigger the liver, the more TIM4neg cells). Therefore, the increase in CD11c cells in the liver could just be a consequence of obesity rather than a cause of the phenotype. In support of the former, the change in weight gain occurs nearly immediately after the initiation of HFD (Fig. 5b), well before TIM4neg/CD11c expressing macrophages should be present in the liver.

5) Although the authors have added additional data about autophagy to complement their LC3-GFP data they didn't really address the issue. Autophagy is a dynamic process and therefore static LC3-GFP quantification, LC3i/LC2ii ratios, and p62 accumulation cannot distinguish increased flux vs. diminished degradation. The data presented here suggest that high fat diet either increases autophagy initiation or decreases lysosomal function (as has been shown to occur with HFD). In order to distinguish these possibilities, the authors would need to assess flux with lysosomal

inhibitors.

Reviewer #2 (Remarks to the Author):

The authors have responded well to most of this reviewer's comments. However, this manuscript still contains a problem in measuring autophagic activity.

In New Fig. 2, the authors provide additional data on autophagic flux. However, the steady-state level of LC3-II (or LC3-II/LC3-I) cannot be used to determine whether autophagic flux is increased or decreased (Reviewer #1 also pointed out this issue). The authors should investigate if LC3-II further accumulates after lysosomal inhibition and compare the difference between NCD and HFD. The protein level of SQSTM1/p62 is regulated by both transcription and autophagic degradation. Therefore, Immunoblotting of SQSTM1/p62 can be useful as far as its transcriptional level is confirmed to be unaffected.

Reviewer #3 (Remarks to the Author):

The authors have satisfactorily addressed my comments. The revised manuscript has been significantly improved and can be considered to be published in Nature Communications.

REVIEWER COMMENTS

Reviewer #1 (Remarks to the Author):

The revised manuscript by Galle-Treger et al is improved and the authors have made a good effort to address the concerns of this reviewer. However, there are still a few issues that should be addressed:

1) The authors were very responsive to my critique that the primary phenotype of the CD11c-ATG5KO is a difference in obesity. Based on their studies, it appears that the KO mice are hyperphagic and eat more food at nighttime leading excessive weight gain. These findings suggest that IL-23 may be acting centrally to regulate satiety centers. I am still not convinced that CD11c cells in the liver are responsible for this phenotype. The authors claim that it can't be related to adipose tissue macrophages because there is no difference in the % of CD11c expressing macrophages between the genotypes. However, the total number of CD11c expressing ATMs is actually much greater in the KO because the perigonadal fat pads are nearly double in size in the KO (and therefore for the same percentage of cells the absolute number of these macrophages per fat pad is significantly increased). Moreover, the amount of IL-23 production in the ATMs from WT and KO mice is not shown. Again, as CD11c-cre deletes throughout the body it cannot be concluded that the hepatic cells are responsible for this phenotype (this is related to point 3 below). How do the authors exclude that increased IL-23 from DCs is not the reason for the phenotype?

We have previously assessed the percentage of CD11c+ macrophages in the visceral adipose tissue (VAT) of mice fed with NCD or HFD (Supplementary figures 1 e,f and 2 a,b). The deletion of Atg5 had no effect on the frequency of the CD11c+ macrophages in the VAT. Based on the reviewer's recommendations, we additionally assessed IL-23 production in VAT CD11c+ macrophages of WT and Atg5 CD11c KO mice fed with HFD for 14 weeks. Atg5 CD11c KO mice fed HFD had a significant increase in the percentage of IL-23+ CD11c+ macrophages in the VAT as compared to WT control mice (Supplementary figure 4f-g). These results demonstrate that in response to HFD, Atg5 deletion increases systemic inflammation and leads to higher IL-23 production by liver and VAT CD11c+ cells. Please note that we have also previously reported that in Atg5 deficient mice the pulmonary CD11c+ cells secreted high levels of IL-23 (PMID 26589586). To confirm the role of IL-23, we demonstrate the results of IL-23 neutralizing antibodies in figures 6-7 of our manuscript. Moreover, recent publications have shown that IL-23 induction was associated with NASH development in other murine models including (Foz/Foz mice) (PMID: 34062281, 24107103).

2) The authors did additional flow cytometry to better quantify the changes in specific macrophage populations with obesity. I am concerned, however, about some of their CLEC4F staining. From several recent papers (PMID: 32888418, 32562600, 33440159), we now know that KCs are depleted and TIM4neg monocyte derived cells enter the liver during NASH development. The TIM4neg cells can be subdivided into Mo-KCs (CLEC4Fpos, VSIG4pos) and "Lipid associated macrophages" referred to as LAMs. The LAMs can also be further subdivided into C-LAMs and LAMs based on the expression of CCR2/Cx3cr1 (PMID:33440159). In general, about 50% of the TIMneg cells are Mo-KCs and the other 50% are LAM subsets. However, in Figure 4 of this manuscript nearly all (96%) of the TIM4neg cells appear to express CLEC4F. This would be extremely surprising. Did the authors also use an isotype control antibody? I am concerned that some of this staining may be non-specific. Based on our own experience, CLEC4F staining by flow can be challenging. This is relevant because it has been shown that C-LAMs and

LAMs have increased CD11c expression and that KCs and Mo-KCs do not (PMID: 32888418, 33440159). Thus, the figure legend title for Fig 4 is misleading and should be changed. In addition, the discussion paragraph about hepatic myeloid cells needs to address this point and should more thoroughly incorporate the findings for the 3 manuscripts referenced above.

We thank the reviewer for their suggestions. We have carefully reanalyzed our data and adapted our gating strategy according to the recommended manuscript (PMID: 32888418). Based on the isotype control of the adjusted gating strategies, we now confirm that TIM4⁺ Clec4f⁺ KC cells represent around 30% of the stromal cells in the liver isolated from HFD fed mice. We now added the new data and discussed the recommended publications in our discussion (PMID: 32888418, 32562600, 33440159). Please note that both figures 3 and 4 were merged into one figure (Figure 3) and the title was updated, as requested by the reviewer. This figure shows clearly now within the same graphs, the effect of the diet as well as Atg5 deletion on CD11c expression and on the distribution of different macrophage subpopulations in the liver.

3) As a more minor point, the box labeled monocytes (Fig 3A, Fig 4A) actually contains both neutrophils and monocytes (monocytes are higher in Ly6C and lower in CD11b, whereas the neutrophils are intermediate Ly6C and higher in CD11b. In addition, the neutrophils are higher SSC. Using these parameters, the authors could re-gate their data to remove the neutrophils.

Following the reviewer's recommendations, we have updated our gating strategy in Figure 3a. Monocytes were gated as live SSC low CD45⁺ Ly6C high and CD11b⁺ cells, this population was clearly separated and identifiable in our dot plots. According to this new gating strategy, neutrophils were excluded from Ly6C⁺ cells based on their high SSC levels. This is now added as supplementary figure 1d.

4) I am still a little unclear about how the authors integrate their data. Since the mice eat more food when fed a HFD they get more obese which leads to more hepatic steatosis and liver injury. This will then cause more LAM infiltration and more KC loss (in our experience the bigger the liver, the more TIM4^{neg} cells). Therefore, the increase in CD11c cells in the liver could just be a consequence of obesity rather than a cause of the phenotype. In support of the former, the change in weight gain occurs nearly immediately after the initiation of HFD (Fig. 5b), well before TIM4^{neg}/CD11c expressing macrophages should be present in the liver.

We apologize for the lack of clarity in our explanation. We agree that diet-induced obesity triggers the development of hepatic steatosis leading to liver injury in our model. This pathogenesis led to increased levels of inflammation and in particular an accumulation of infiltrating LAM and loss of KC. Consistently, our results in Figure 3c clearly show that mice fed HFD regardless of their genotype have decreased liver TIM4⁺ KC percentages whereas TIM4⁻ KC frequencies are increased. Interestingly the percentage of CD11⁺ DC is specifically increased in the liver of Atg5 CD11c KO mice fed HFD. We would like to highlight however that even though the weight gain is at the origin of the inflammation, the treatment with the neutralizing anti-IL-23 antibody resolved the inflammation and the liver steatosis in Atg5 CD11c KO mice fed HFD without affecting their weight. These data clearly establish that IL-23 is the key to improving the inflammation status and the liver injury in our model (Figures 6 and 7).

5) Although the authors have added additional data about autophagy to complement their LC3-GFP data they didn't really address the issue. Autophagy is a dynamic process and therefore static LC3-GFP quantification, LC3i/LC2ii ratios, and p62 accumulation cannot distinguish increased flux vs. diminished

degradation. The data presented here suggest that high fat diet either increases autophagy initiation or decreases lysosomal function (as has been shown to occur with HFD). In order to distinguish these possibilities, the authors would need to assess flux with lysosomal inhibitors.

We understand the reviewer's comments and we know that multiple defects could lead to autophagy dysfunction in our model. The early phase of the autophagy mechanism, in particular, the induction and the autophagosome formation are key steps that could affect this mechanism of action and similarly, the inefficient degradation of the autophagic vesicles in the lysosome could also be involved.

To complete our previous set of results, we first assessed the LC3-II/LC3-I conversion ratio and p62 accumulation in hepatic CD11c+ cells of WT mice fed NCD and HFD.

In supplementary figure 1a-b, we isolated hepatic CD11c+ cells from WT mice fed NCD and HFD, and treated the cells with chloroquine (CQ), an autophagy inhibitor that blocks the fusion of autophagosomes to lysosomes by altering the acidic environment of lysosomes. Subsequently, CQ blocks p62 and LC3 degradation. Our results show that in the presence of CQ, we have an impaired LC3-I to LC3-II conversion and an aberrant p62 accumulation in the hepatic CD11c+ cells from WT mice fed HFD compared to WT mice fed NCD, indicating a disruption to lysosomal degradation. To confirm the effect of HFD on LC3 degradation, we treated liver CD11c+ cells isolated from LC3-GFP mice fed NCD and HFD with CQ. Without the CQ treatment, hepatic CD11c+ cells isolated from HFD mice have a baseline accumulation of total LC3 protein compared to cells isolated from NCD mice, showing impairment of autophagy in response to HFD at a steady state. Interestingly, the total LC3 protein levels were even higher after CQ treatment confirming the impaired degradation (Figure 2c). Altogether these results establish that autophagic flux is impaired in mice when fed HFD, leading to diminished degradation. We would also like to emphasize that on average we isolate 40,000 CD11c+ myeloid cells in the liver of mice fed NCD against 60,000 cells in the liver of mice fed HFD. To run a western blot, each sample represents the CD11c+ cells isolated in the livers of 25-30 mice fed HFD for 2-3 months. These experiments performed in primary cells are technically challenging due to the low yield of these cell populations, thus highlighting the value of these results.

Reviewer #2 (Remarks to the Author):

The authors have responded well to most of this reviewer's comments. However, this manuscript still contains a problem in measuring autophagic activity.

In New Fig. 2, the authors provide additional data on autophagic flux. However, the steady-state level of LC3-II (or LC3-II/LC3-I) cannot be used to determine whether autophagic flux is increased or decreased (Reviewer #1 also pointed out this issue). The authors should investigate if LC3-II further accumulates after lysosomal inhibition and compare the difference between NCD and HFD. The protein level of SQSTM1/p62 is regulated by both transcription and autophagic degradation. Therefore, Immunoblotting of SQSTM1/p62 can be useful as far as its transcriptional level is confirmed to be unaffected.

As we previously explained for reviewer#1 question 5, we have performed the recommended experiments and assessed LC3-II and p62 accumulation in presence of chloroquine in hepatic CD11c+ cells in figures 2b-c and supplementary figure 1a-b. Our Western blot results show that in the presence of CQ, we have an impaired LC3-I to LC3-II conversion and an aberrant p62 accumulation in the hepatic CD11c+ cells from WT mice fed HFD compared to WT mice fed NCD, indicating a disruption to lysosomal degradation. To confirm the effect of HFD on LC3 degradation, we treated liver CD11c+ cells isolated from LC3-GFP mice

fed NCD and HFD with CQ. Without the CQ treatment, hepatic CD11c+ cells isolated from HFD mice have a baseline accumulation of total LC3 protein compared to cells isolated from NCD mice, showing impairment of autophagy in response to HFD at a steady state. Interestingly, the total LC3 protein levels were even higher after CQ treatment confirming the impaired degradation (Figure 2c). Altogether these results establish that autophagic flux is impaired in mice when fed HFD, leading to diminished degradation.

Reviewer #3 (Remarks to the Author):

The authors have satisfactorily addressed my comments. The revised manuscript has been significantly improved and can be considered to be published in Nature Communications.

We would like to thank reviewer 3 for the positive comments and recommendations.

REVIEWERS' COMMENTS

Reviewer #1 (Remarks to the Author):

The authors have performed additional experiments and re-analyzed their data based on my recommendations. The revised manuscript is much improved. I have no further comments.

Reviewer #2 (Remarks to the Author):

The authors have now included data of autophagic flux in Supplementary Figure 1a and b, which supports the authors' hypothesis that autophagy is impaired in HFD mice. This is consistent with the previous data in Figure 2b and new data in Figure 2C. Ideally speaking, there are still several issues that would be better to addressed. For example, it is unclear whether the high expression level of p62 in HFD cells is indeed due to a defect in degradation rather than transcriptional upregulation (Figure 2C; a question that this reviewer asked in the previous round of review), and whether the new important data in Supplementary Figure 1a-b are reproducible (only n=1 result is shown). In addition, the authors previously showed that the GFP-LC3 intensity was lower in HFD cells than in control cells (previous Figure 2b), but now they show the opposite results (new Figure 2c) without providing reasonable explanations. However, even with these issues, this reviewer thinks that the authors' conclusion is now mostly supported by the provided data.

Reviewer #1 (Remarks to the Author):

The authors have performed additional experiments and re-analyzed their data based on my recommendations. The revised manuscript is much improved. I have no further comments.

We would like to thank reviewer 1 for their constructive comments.

Reviewer #2 (Remarks to the Author):

The authors have now included data of autophagic flux in Supplementary Figure 1a and b, which supports the authors' hypothesis that autophagy is impaired in HFD mice. This is consistent with the previous data in Figure 2b and new data in Figure 2C. Ideally speaking, there are still several issues that would be better to address. For example, it is unclear whether the high expression level of p62 in HFD cells is indeed due to a defect in degradation rather than transcriptional upregulation (Figure 2C; a question that this reviewer asked in the previous round of review), and whether the new important data in Supplementary Figure 1a-b are reproducible (only n=1 result is shown). In addition, the authors previously showed that the GFP-LC3 intensity was lower in HFD cells than in control cells (previous Figure 2b), but now they show the opposite results (new Figure 2c) without providing reasonable explanations. However, even with these issues, this reviewer thinks that the authors' conclusion is now mostly supported by the provided data.

We thank the reviewer for their positive remarks. We need to point out that we have assessed LC3-II and p62 accumulation by western blot several times and during the revision we performed one more time in the presence of chloroquine and other in vitro treatments as a proof of concept and to reconfirm our previous results. It is also important to point out that we did not only rely on western blot, but also confirmed our results by flow cytometry. Flow cytometry data in Figure 2c provided an accurate measurement of total LC3 accumulation and thus replace immunofluorescence microscopy (previous Figure 2b) that only allows a semi-quantitative detection of LC3 and could be affected by different sources of noise including autofluorescence. Furthermore, our results were further supported by our transcriptome analysis in both human and mice livers presented in Figure 1. The data in Figure 1 do not show an upregulation in the transcript levels of p62 encoding gene (SQSTM1 and Sqstm1) in both human and mice livers, and as matter of fact showed some degree of downregulation. This observation suggests that p62 accumulation in response to HFD is due to a defect in protein degradation rather than transcriptional upregulation, and therefore supports our previous conclusions. We now highlighted these results in our revised manuscript to clarify these important points for the readers of *Nature Communications*.

We would like to emphasize (as we stated in our previous point-by-point response) that western blot experiments on rare but important primary cells (tissue-resident CD11c+ cells) are technically challenging. For that reason, we designed several different approaches to reach confidence in the results obtained.

Finally, we are glad that all the three reviewers agree that our results, targeting these previously unappreciated innate cells and pathways such as IL-23, might provide an important treatment option for patients suffering from obesity and fatty liver disease- opening an avenue for

further analyzing and designing targeted therapies. Obviously, how exactly autophagy deficiency affects IL-23 signaling is beyond the scope of this paper and remains to be explored.